# Perceptual Kalman Filters: Online State Estimation under a Perfect Perceptual-Quality Constraint

**Dror Freirich**
Technion - Israel
Institute of Technology
drorfrc@gmail.com

**Tomer Michaeli**
Technion - Israel
Institute of Technology
tomer.m@ee.technion.ac.il

**Ron Meir**
Technion - Israel
Institute of Technology
rmeir@ee.technion.ac.il

## Abstract

Many practical settings call for the reconstruction of temporal signals from corrupted or missing data. Classic examples include decoding, tracking, signal enhancement and denoising. Since the reconstructed signals are ultimately viewed by humans, it is desirable to achieve reconstructions that are pleasing to human perception. Mathematically, perfect perceptual-quality is achieved when the distribution of restored signals is the same as that of natural signals, a requirement which has been heavily researched in static estimation settings (i.e. when a whole signal is processed at once). Here, we study the problem of optimal *causal* filtering under a perfect perceptual-quality constraint, which is a task of fundamentally different nature. Specifically, we analyze a Gaussian Markov signal observed through a linear noisy transformation. In the absence of perceptual constraints, the Kalman filter is known to be optimal in the MSE sense for this setting. Here, we show that adding the perfect perceptual quality constraint (i.e. the requirement of temporal consistency), introduces a fundamental dilemma whereby the filter may have to "knowingly" ignore new information revealed by the observations in order to conform to its past decisions. This often comes at the cost of a significant increase in the MSE (beyond that encountered in static settings). Our analysis goes beyond the classic innovation process of the Kalman filter, and introduces the novel concept of an unutilized information process. Using this tool, we present a recursive formula for perceptual filters, and demonstrate the qualitative effects of perfect perceptual-quality estimation on a video reconstruction problem.

## 1 Introduction

In many settings, it is desired to reconstruct a temporal signal from corrupted or missing data. Examples include decoding of transmitted communications, tracking targets based on noisy measurements, enhancing audio signals, and denoising videos. Traditionally, restoration quality has been assessed by distortion measures such as MSE. As a result, numerous methods targeted the minimization of such measures, including the seminal work of Kalman [11]. However, in applications involving human perception, one may favor reconstructions that cannot be told apart from valid signals. Mathematically, such *perfect perceptual quality* can be achieved only if the distribution of restored signals is the same as that of "natural" signals.

Interestingly, it has been shown that good perceptual quality generally comes at the price of poor distortion and vice versa. This phenomenon, known as the *perception-distortion tradeoff*, was first studied in [3], and was later fully characterized in [8] for the particular setting where distortion is measured by MSE and perception is measured by the Wasserstein-2 distance between the distribution of estimated signals and the distribution of real signals. However, to date, all existing works addressed the static (non-temporal) setting, in which the entire corrupted signal is available for processing all at

37th Conference on Neural Information Processing Systems (NeurIPS 2023).

once. This setting is fundamentally different from situations involving temporal signals, in which the corrupted signal is processed causally over time, such that each sample is reconstructed only based on observations up to the current time.

To illustrate the inherent difficulty in causal estimation, consider video restoration tasks like denoising, super-resolution, or frame completion (see Fig. 1). Achieving high perceptual quality in those tasks requires generating restored videos whose spatio-temporal distribution matches that of natural videos. Particularly, an incorrect temporal distribution may lead to flickering artifacts [14] or to unnaturally slow dynamics [4]. To comply with this requirement, the restoration method needs to 'hallucinate' motion whenever the dynamics cannot be accurately determined from the measurements. For example, it may be impossible to determine whether a car is standing still or moving slowly from just a few noisy frames, yet the restoration method must generate *some* (nearly) constant velocity in order to comply with the statistics of natural videos. However, as more measurements become avail-

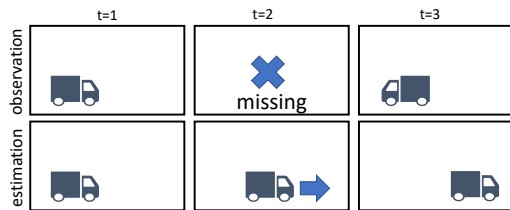

Figure 1: **The temporal consistency dilemma.** Estimation cannot suddenly change the motion in the output video, because such an abrupt change would deviate from natural video statistics. Thus, although the method is aware of its mistake, it may have to stick to its past decisions.

able, the uncertainty may be reduced, and it may become evident that the hallucinated dynamics were in fact incorrect. When this happens, the method cannot suddenly change the motion in the output video, because such an abrupt change would deviate from natural video statistics. Thus, although the method "becomes aware" of its mistake, it may have to stick to its past decisions for a while. A natural question is, therefore:

*What is the precise cost of temporal consistency in online restoration?*

In this paper, we study this question in the setting where the signal to be restored, $x_t$, is a discrete-time Gaussian Markov process, and the measurements $y_t$ are noisy linear transformations of the signal's state. We address the problem of designing a *causal filter* for estimating $x_t$ from $y_t$, where the distribution law of the filter's output, $\hat{x}_t$, is constrained to be the same as that of $x_t$ (perfect perceptual quality). We show that this temporal consistency constraint indeed comes at the cost of increased MSE compared to filters that only enforce the correct distribution per time step, but not joint distributions across time steps. To derive a recursive form for linear perceptual filters, we introduce the novel concept of an *unutilized information* process, which is the portion of accumulated information that does not depend on past estimates. We provide a closed-form expression for the MSE of such filters and show how to design their coefficients to minimize different objectives. We further establish a special class of perceptual filters, based on the classic *innovation* process, which has an explicit solution. We analyze the evolution of MSE over time for perceptual filters and for non-perceptual ones in several numerical setups. Finally, we demonstrate the qualitative effects of perfect perceptual-quality estimation on a simplified video reconstruction problem.

**Related work**  Many works proposed practical algorithms for achieving high (spatio-temporal) perceptual quality in video restoration tasks. Bhattacharjee and Das [1] improved temporal coherence by using a loss that penalizes discontinuities between sequential frames. Pérez-Pellitero et al. [14] suggested a recurrent generator architecture whose inputs include the low-resolution current frame (at time $t$), the high-resolution reconstruction of the previous one (at time $t - 1$), and a low-resolution version of the previous frame, aligned to the current one. The model is trained using losses that encourage it to conserve the joint statistics between consecutive frames. Chu et al. [4] introduced a temporally coherent GAN architecture (TecoGAN). Their generator's input includes again a warped version of the previously generated frame, where each discriminator input consists of 3 consecutive frames, either generated or ground-truth. They also introduced a bi-directional loss that encourages long-term consistency by avoiding temporal accumulation of artifacts, and another loss which measures the similarity between motions. More recent progress in generating temporally coherent videos includes *Make-a-video* [16] that expands a text-to image model with spatio-temporal convolutional and attention layers, and Video-Diffusion Models [10] that use a

diffusion model with a network architecture adapted to video. In the context of online restoration, we mention the work of Kim et al. [12] which presented a GAN architecture for real-time video deblurring, where restoration is done sequentially. They introduce a network layer for dynamically (at test-time) blending features between consecutive frames. This mechanism enables generated features to propagate into future frames, thus improving consistency. We note, however, that our work is the first to provide a theoretical/mathematical framework, and a closed-form solution for a special case.

## 2 Preliminaries: The distortion-perception tradeoff

Let $x, y$ be random vectors taking values in $\mathbb{R}^{n_x}$ and $\mathbb{R}^{n_y}$, respectively, with joint probability $p_{xy}$. Suppose we want to estimate $x$ based on $y$, such that the estimator $\hat{x}$ satisfies two requirements: (i) It has a low distortion $\mathbb{E}[d(x, \hat{x})]$, where $d(\cdot, \cdot)$ is some measure of discrepancy between signals; (ii) It has a good perceptual quality, *i.e.* it achieves a low value of $d_p(p_x, p_{\hat{x}})$, where $d_p(\cdot, \cdot)$ is a divergence between probability measures. Blau and Michaeli [3] studied the best possible distortion that can be achieved under a given level of perceptual quality, by introducing the *distortion-perception* function

$$D(P) = \min_{p_{\hat{x}|y}} \{ \mathbb{E}[d(x, \hat{x})] \ : \ d_p(p_x, p_{\hat{x}}) \leq P \}. \tag{1}$$

Freirich et al. [8] provided a complete characterization of $D(P)$ for the case where $d$ is the squared-error and $d_p$ is the Wasserstein-2 distance. Particularly, in the Gaussian case, they developed a closed-form expression for the optimal estimator.

In this paper we discuss estimation with perfect perceptual quality, namely $P = 0$. In this case, [8, Thm. 4] implies that if $x$ and $y$ are zero-mean, jointly-Gaussian with covariances $\Sigma_x, \Sigma_y \succ 0$, and $x^* = \mathbb{E}[x|y]$, then a MSE-optimal perfect perceptual-quality estimator is obtained by

$$\hat{x} = \mathscr{T}^* x^* + w, \quad \mathscr{T}^* \triangleq \Sigma_x^{\frac{1}{2}} (\Sigma_x^{\frac{1}{2}} \Sigma_{x^*} \Sigma_x^{\frac{1}{2}})^{\frac{1}{2}} \Sigma_x^{-\frac{1}{2}} \Sigma_{x^*}^{\dagger}, \tag{2}$$

where $w$ is a zero-mean Gaussian noise with covariance $\Sigma_w = \Sigma_x - \mathscr{T}^* \Sigma_{x^*} \mathscr{T}^{*\top}$, independent of $y$ and $x$, and $\Sigma_{x^*}^{\dagger}$ is the Moore-Penrose inverse of $\Sigma_{x^*}$. For the more general case where $\Sigma_x \succeq 0$, a similar result can be obtained by using Theorem F.2 in the Appendix.

## 3 Problem formulation

We consider a state $x_k \in \mathbb{R}^{n_x}$ with linear dynamics driven by Gaussian noise, and observations $y_k \in \mathbb{R}^{n_y}$ that are linear transformations of $x_k$ perturbed by Gaussian noise,

$$x_k = A_k x_{k-1} + q_k, \qquad\qquad k = 1, ..., T, \tag{3}$$
$$y_k = C_k x_k + r_k, \qquad\qquad k = 0, ..., T. \tag{4}$$

Here, the noise vectors $q_k \sim \mathcal{N}(0, Q_k)$ and $r_k \sim \mathcal{N}(0, R_k)$ are independent white Gaussian processes, and $x_0 \sim \mathcal{N}(0, P_0)$ is independent of $q_1, r_0$. For convenience, we will sometimes refer to $P_0$ as $Q_0$. The matrices $A_k, C_k, Q_k, R_k$ and $P_0$ are deterministic system parameters with appropriate dimensions, and assumed to be known.

Our goal is to construct an estimated sequence $\hat{X}_0^T = (\hat{x}_0, \ldots, \hat{x}_T)$ based on the measurements $Y_0^T = (y_0, \ldots, y_T)$, which minimizes the cost

$$\mathcal{C}(\hat{x}_0, \ldots, \hat{x}_T) = \sum_{k=0}^{T} \alpha_k \mathbb{E}\left[ \|x_k - \hat{x}_k\|^2 \right], \tag{5}$$

for some given weights $\alpha_k \geq 0$. Importantly, we want to do so under the following two constraints.

$$\text{Temporal causality}: \qquad \hat{x}_k \sim p_{\hat{x}_k}(\cdot | y_0, \ldots, y_k, \hat{x}_0, \ldots, \hat{x}_{k-1}), \tag{6}$$
$$\text{Perfect perceptual quality}: \qquad p_{\hat{X}_0^T} = p_{X_0^T}. \tag{7}$$

Condition (6) states that each prediction should depend only on past and present observations and on the past predictions. Note that Condition (7) requires not only that every estimated sample have the same distribution as the original one, but also that the *joint* distribution of every subset of reconstructed samples be identical to that of the corresponding subset of samples in the original sequence. In the context of video processing, this means that not only does every recovered frame have to look natural, but also that motion must look natural. This perfect perceptual quality constraint is what sets our problem apart from the classical Kalman filtering problem, which considers only the causality constraint. Since we will make use of the Kalman filter, let us briefly summarize it.

**The Kalman filter (no perceptual quality constraint)** Let $\hat{x}^*_{k|s} \triangleq \mathbb{E}\left[x_k|y_0,\ldots,y_s\right]$ denote the estimator of $x_k$ based on all observations up to time $s$, which minimizes the MSE. The celebrated Kalman filter [11] is an efficient method for calculating the *Kalman optimal state* $\hat{x}^*_k \equiv \hat{x}^*_{k|k}$ recursively without having to store all observations up to time $k$. It is given by the recurrence

$$\hat{x}^*_k = A_k\hat{x}^*_{k-1} + K_k\mathcal{I}_k, \tag{8}$$

where $K_k$ is the *optimal Kalman gain* [11], whose recursive calculation is given in Algo. 2 in the Appendix. The vector $\mathcal{I}_k$ is the *innovation* process,

$$\mathcal{I}_k = y_k - C_k\hat{x}^*_{k|k-1}, \tag{9}$$

describing the new information carried by the observation $y_k$ over the optimal prediction based on the observations up to time $k-1$, which is given by $\hat{x}^*_{k|k-1} = A_k\hat{x}^*_{k-1}$. The innovation $\mathcal{I}_k$ is uncorrelated with all observations up to time $k-1$, which guaranties the MSE optimality of the estimation. The calculation of the Kalman state is also summarized in Alg. 2. Pay attention to the innovation process $\mathcal{I}_k$, its covariance $S_k$ and gain $K_k$, which we will build upon. Our notations are summarized in Table 2 in the Appendix. Note that since the Kalman filter minimizes the MSE at each timestep, it also minimizes (5) regardless of the choice of $\alpha_k$, but it generally fails to fulfill (7). As we will see later, when taking (7) into consideration, the choice of $\alpha_k$ does affect the optimal filter.

**Temporally-inconsistent perceptual filter** A naive way to try to improve the perceptual quality of the Kalman filter would be to require that each $\hat{x}_k$ be distributed like $x_k$ (but without constraining the joint distribution of samples). In the context of video processing, each frame generated by such a filter would look natural, but motion would not necessarily look natural. This problem can be solved using the result (2), which gives the optimal "temporally-inconsistent" perceptual estimator

$$\hat{x}^{\text{tic}}_k = \mathscr{T}^*_k\hat{x}^*_k + w_k = \mathscr{T}^*_k\left(A_k\hat{x}^*_{k-1} + K_k\mathcal{I}_k\right) + w_k, \tag{10}$$

with $\mathscr{T}^*_k$ and $w_k$ from (2). These quantities depend only on the covariances of $x_k, \hat{x}^*_k$, which can be computed recursively using the Kalman method. The MSE of this estimator is given by (see [8])

$$\mathbb{E}\left[\|x_k - \hat{x}_k\|^2\right] = d^*_k + \text{Tr}\left\{\Sigma_{x_k} + \Sigma_{\hat{x}^*_k} - 2\left(\Sigma^{\frac{1}{2}}_{x_k}\Sigma_{\hat{x}^*_k}\Sigma^{\frac{1}{2}}_{x_k}\right)^{\frac{1}{2}}\right\}, \tag{11}$$

where $d^*_k$ is the MSE of the Kalman filter, which can also be computed recursively.

**Our setting (with the perceptual quality constraint)** Going back to our setting, one may readily recognize that perceptually reconstructing the signal $X^T_0$ from the full measurement sequence $Y^T_0$ is also a special case of the Gaussian perceptual restoration problem discussed in Section 2, only applied to the entire sequence of states and measurements. Generally, this estimate already achieves a higher MSE than the estimate that minimizes the MSE without the perceptual constraint. However, in our setting we have the additional causality constraint (6). Requiring both constraints (7) and (6) might incur an additional cost, as illustrated by the following example, where applying each one of them does not restrict the optimal solution, but together they result in a higher MSE.

*Example* 3.1. Let $T = 1$ and consider the process $(x_0, x_1) = (q_0, q_0)$, where $q_0 \sim \mathcal{N}(0,1)$, with observations $(y_0, y_1) = (0, x_1)$. Assume we want to minimize the error at time $k = 1$ (namely $(\alpha_0, \alpha_1) = (0, 1)$ in (5)). Then, considering only the causality constraint (6), the estimator $(\hat{x}_0, \hat{x}_1) = (y_0, y_1)$ is optimal. Indeed, it is causal and it achieves zero MSE. Similarly, considering only the perceptual quality constraint (7), the estimator $(\hat{x}_0, \hat{x}_1) = (y_1, y_1)$ is optimal. Indeed, it is distributed like $(x_0, x_1)$ and it also achieves zero MSE. However, when demanding both conditions, $\hat{x}_0$ must be based on no information to obey (6) (as $y_0 = 0$), and it must be drawn from the prior distribution $\mathcal{N}(0,1)$ in order to be distributed like $x_0$ and obey (7). Furthermore, to satisfy (7), we must also have $\hat{x}_1 = \hat{x}_0$. Therefore, the optimal estimator in this case is $(\hat{x}_0, \hat{x}_1) = (\tilde{q}_0, \tilde{q}_0)$, where $\tilde{q}_0 \sim \mathcal{N}(0,1)$ is independent of $q_0$. The MSE achieved by this estimator is 2.

## 4   Perfect perceptual-quality filters

The perceptual constraint (7) dictates that the estimator must be of the form

$$\hat{x}_k = A_k\hat{x}_{k-1} + J_k, \quad \hat{x}_0 = J_0, \tag{12}$$

where $J_k = \hat{x}_k - A_k\hat{x}_{k-1}$ is distributed as $\mathcal{N}(0, Q_k)$ and is independent of $\hat{X}_0^{k-1}$. Note the similarity between (12) and the MSE-optimal state (8), in which $J_k = K_k\mathcal{I}_k$. Here, however, this choice is not valid due to the constraint on the output distribution. In terms of temporal consistency, an estimator of the form (12) guarantees that previously presented features obey the natural dynamics of the domain, while newly generated estimates do not contradict the previous ones. In order to maintain causality (6), $J_k$ must be of the form

$$J_k \sim p_{J_k}(\cdot|y_0, \ldots, y_k, \hat{x}_0, \ldots, \hat{x}_{k-1}), \tag{13}$$

*i.e.*, $J_k$ is independent of future observations $Y_{k+1}^T$ given $Y_0^k$. As a consequence, $J_k$ is uncorrelated with $\mathcal{I}_{k+n}$ for all $n \geq 1$.

We now discuss linear estimators, where $Y_0^T$ and $J_0^T$ (hence $\hat{X}_0^T$) are jointly Gaussian. Our first result is as follows (see proof in App. B).

**Theorem 4.1.** *Under the cost* (5)*, there exists a linear optimal estimator of the form*

$$J_k = \pi_k\mathcal{I}_k + \phi_k\upsilon_k + w_k, \tag{14}$$

$\pi_k \in \mathbb{R}^{n_x \times n_y}$ *and* $\phi_k \in \mathbb{R}^{n_x \times (kn_y)}$ *are the filter's coefficients,* $w_k$ *is an independent Gaussian noise with covariance* $\Sigma_{w_k} = Q_k - \pi_k S_k \pi_k^\top - \phi_k \Sigma_{\upsilon_k} \phi_k^\top \succeq 0$, $\upsilon_k$ *is the process of* unutilized information

$$\upsilon_k \triangleq \mathcal{I}_0^{k-1} - \mathbb{E}\left[\mathcal{I}_0^{k-1}\middle|\hat{X}_0^{k-1}\right], \; \mathcal{I}_0^{k-1} = \text{Column}\left\{\mathcal{I}_0, \ldots, \mathcal{I}_{k-1}\right\}, \upsilon_0 = 0. \tag{15}$$

Note that with this form for $J_k$, the state $\hat{x}_k$ is indeed a function of the observations $Y_0^k$ and the previous states $\hat{X}_0^{k-1}$. Intuitively, $\upsilon_k$ is the part of the information in the observations, which has no correlation with the information used to construct the past estimates $\hat{X}_0^{k-1}$. Thus, from the standpoint of the filter's output, this information has not yet been introduced. As opposed to the innovation $\mathcal{I}_k$, the process $\upsilon_k$ is not white, and it is affected by the choices of $\pi_t$ and $\phi_t$ up to time $k - 1$. However, $\mathcal{I}_k$ is always independent of $\upsilon_k$, since $\mathcal{I}_k$ is independent of $\mathcal{I}_0^{k-1}$ and $J_0^{k-1}$, which constitute $\upsilon_k$.

The filter of Theorem 4.1 is causal but not recursive. Specifically, although it is possible to obtain $\upsilon_{k+1}, \Sigma_{\upsilon_{k+1}}$ given the coefficients $\{\pi_t, \phi_t\}_{t=0}^k$ (see App. E), the dimension of $\upsilon_k$ grows with time (it is $kn_y$), thus increasing the cost of computing $\phi_k\upsilon_k$. Furthermore, determining the coefficients $\{\pi_k, \phi_k\}_{k=0}^T$ that minimize the objective (5) (which is done a-priori in an offline manner) requires solving a large optimization problem, as the total size of all coefficients is $\mathcal{O}(n_x n_y T^2)$. To efficiently optimize these coefficients, we next suggest two simplified versions of this form, which may generally be sub-optimal but easier to optimize.

*Remark* 4.2. A remark is in place regarding objectives beyond the

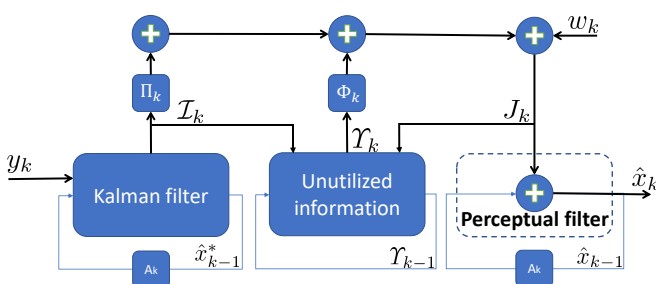

Figure 2: **Recursive perceptual filtering (Sec. 4.1).** The state estimator $\hat{x}_k$ consists of the previous state $\hat{x}_{k-1}$, and the innovation and unutilized information processes. The unutilized information state $\Upsilon_{k+1}$ is updated using the previously unutilized information $\Upsilon_k$ and the newly-arriving information $\mathcal{I}_k$. The currently utilized information, arriving from $J_k$, is then subtracted from $\Upsilon_{k+1}$.

squared-error. While (14) forms an optimal filter under the cost (5), it can be considered as a representation for linear filters in general. The constraints on the coefficients $(Q_k - \pi_k S_k \pi_k^\top - \phi_k \Sigma_{\upsilon_k} \phi_k^\top \succeq 0)$ are necessary and sufficient for perfect perception, regardless of the cost objective. It is therefore possible to optimize coefficients for objectives other than MSE under these constraints to obtain optimal perceptual *linear* filters. Optimization may be performed numerically (when the cost is tractable) or in an online fashion, assuming access to ground-truth samples. Note, however, that under general distortion measures, optimal filters might be non-linear.

## 4.1 Recursive-form filters

The optimal Kalman state $\hat{x}_k^*$ achieves the minimal possible MSE, given by $d_k^* = \mathbb{E}\left[\|\hat{x}_k^* - x_k\|^2\right] = \mathrm{Tr}\left\{P_{k|k}\right\}$, where $P_{k|k}$ is the error covariance, given explicitly in Alg. 2. By the orthogonality principle (see *e.g.* [8, Lemma 2]), any other estimator $\hat{x}_k$ based on the observations $Y_0^k$, satisfies

$$\mathbb{E}\left[\|x_k - \hat{x}_k\|^2\right] = d_k^* + \mathbb{E}\left[\|\hat{x}_k - \hat{x}_k^*\|^2\right]. \tag{16}$$

Now, consider an estimator $\hat{x}_k$ of the form (12), and let $D_k \triangleq \mathbb{E}[(\hat{x}_k^* - \hat{x}_k)(\hat{x}_k^* - \hat{x}_k)^\top]$. Since we choose $J_k$ to be normally distributed and independent of $\hat{x}_{k-1}$, it is easy to see that $D_k$ obeys the *Lyapunov difference equation*

$$\begin{aligned}
D_k = &A_k D_{k-1} A_k^\top + K_k S_k K_k^\top + Q_k \\
&- \mathbb{E}[J_k \mathcal{I}_k^\top] K_k^\top - K_k \mathbb{E}[\mathcal{I}_k J_k^\top] - A_k \mathbb{E}[\hat{x}_{k-1}^* J_k^\top] - \mathbb{E}[J_k \hat{x}_{k-1}^{*\top}] A_k^\top. 
\end{aligned} \tag{17}$$

As we see, the choice of $J_k$ affects current (and future) errors by its correlation with the two independent components, $(A_k \hat{x}_{k-1}^*, K_k \mathcal{I}_k)$. Let us now consider filters of the form

$$J_k = \Phi_k A_k \Upsilon_k + \Pi_k K_k \mathcal{I}_k + w_k, \quad w_k \sim \mathcal{N}\left(0, \Sigma_{w_k}\right), \tag{18}$$

where here, by slight abuse of notation, we define the process of unutilized information as

$$\Upsilon_k \triangleq \hat{x}_{k-1}^* - \mathbb{E}\left[\hat{x}_{k-1}^* | \hat{x}_0, \dots, \hat{x}_{k-1}\right], \quad \Upsilon_0 = 0. \tag{19}$$

The matrices $\Pi_k, \Phi_k \in \mathbb{R}^{n_x \times n_x}$ are coefficients such that

$$\Sigma_{w_k} = Q_k - \Phi_k A_k \Sigma_{\Upsilon_k} A_k^\top \Phi_k^\top - \Pi_k M_k \Pi_k^\top \succeq 0, \tag{20}$$

where we denote the Kalman update covariance by $M_k \triangleq K_k S_k K_k^\top$. This guarantees that $J_k \sim N(0, Q_k)$, as desired. Importantly, as opposed to $v_k$, the dimension of $\Upsilon_k$ is *fixed*, namely it does not grow with time $k$. Note that since $\hat{x}_{k-1}^*$ is a linear combination of $(\mathcal{I}_0, \dots, \mathcal{I}_{k-1})$, (18) is a special choice of $\pi_k$ and $\phi_k$ in (14) where coefficient size does not grow with $k$ as well. $\Upsilon_k$ and its covariance $\Sigma_{\Upsilon_k}$ are given via a recursive form, illustrated in Fig. 2 (and derived in App. E):

$$\Upsilon_{k+1} = A_k \Upsilon_k + K_k \mathcal{I}_k - \Psi_k Q_k^\dagger J_k, \ \Sigma_{\Upsilon_{k+1}} = A_k \Sigma_{\Upsilon_k} A_k^\top + M_k - \Psi_k Q_k^\dagger \Psi_k^\top, \tag{21}$$

$$\Psi_k \triangleq M_k \Pi_k^\top + A_k \Sigma_{\Upsilon_k} A_k^\top \Phi_k^\top. \tag{22}$$

Note again that unlike the innovation $\mathcal{I}_k$, $\Upsilon_k$ might not be a white process, but we have that $\Upsilon_k$ is independent of the filter's output $\hat{X}_0^{k-1}$ and $\mathcal{I}_k$. Equation (17) now takes the form

$$D_k = A_k D_{k-1} A_k^\top + Q_k + M_k - \Pi_k M_k - M_k \Pi_k^\top - A_k \Sigma_{\Upsilon_k} A_k^\top \Phi_k^\top - \Phi_k A_k \Sigma_{\Upsilon_k} A_k^\top, \tag{23}$$

where we observe that $\Sigma_{\Upsilon_k}$ may depend on the choice of $\{\Pi_t, \Phi_t\}_{t=0}^{k-1}$. In order to retrieve an optimal filter, one should perform optimization of the desired objective over $\{\Pi_t, \Phi_t\}_{t=0}^T$, under the constraints given in (20). From (16), minimizing the cost (5) boils down to minimizing $\sum_{k=0}^T \alpha_k \mathrm{Tr}\left\{D_k\right\}$ subject to the constraints in (20), which is an optimization problem over only $\mathcal{O}(n_x^2 T)$ parameters.

## 4.2 An Exactly solvable reduction: Perceptual Kalman Filter

We now consider an additional reduction, which allows to obtain a closed form solution for the filter's coefficients. Specifically, a reduced-size filter can be obtained by using the form (12) and (18) with the sub-optimal choice $\Phi_k \equiv 0$, namely

$$J_k = \Pi_k K_k \mathcal{I}_k + w_k. \tag{24}$$

The meaning of this choice is that only newly-observed information is used for updating estimation at each stage, while non-utilized information from previous time steps is discarded. We note that such a simplification should be used with discretion; while requiring only half of the computations, rejecting the unutilized information might lead to enhanced errors in some settings (*e.g.* when an observation is missing). However, in many cases where observations are informative and different timesteps are weakly correlated, past unutilized information rapidly becomes irrelevant and can be safely ignored. We demonstrate the utility of this reduction in Sec. 5. Here, $\Pi_k$ is a $n_x \times n_x$ coefficient matrix, and

---

**Algorithm 1** Perceptual Kalman Filter (PKF)

---

**Input**: Kalman Filter outputs $\{\mathcal{I}_k, K_k, S_k\}_{k=0}^T$, weights $\{\alpha_k\}_{k=0}^T$, matrices $A, P_0, \{Q_k\}_{k=1}^T$ satisfying $\text{Im}\{B_k M_k B_k\} \subseteq \text{Im}\{Q_k\}$ for all $k = 0\dots, T$.

**initialize**: $\hat{x}_0 = \Pi_0 K_0 y_0 + w_0$, $w_0 \sim \mathcal{N}\left(0, P_0 - \Pi_0 M_0 \Pi_0^\top\right)$, where $\Pi_0$ is given by (30) and $B_0 = \sum_{t=0}^T \alpha_t (A^t)^\top A^t$.

**for** $k = 1$ **to** $T$ **do**

    **calculate**: $M_k = K_k S_k K_k^\top$, $B_k = \sum_{t=k}^T \alpha_t (A^{t-k})^\top A^{t-k}$, $M_B = B_k M_k B_k$.

    **compute optimal gain** (30): $\Pi_k = Q_k M_B^{\frac{1}{2}} \left(M_B^{\frac{1}{2}} Q_k M_B^{\frac{1}{2}}\right)^{\frac{1}{2}\dagger} M_B^{\dagger \frac{1}{2}} B_k M_k M_k^\dagger$.

    **sample**: $w_k \sim \mathcal{N}\left(0, Q_k - \Pi_k M_k \Pi_k^\top\right)$.

    **update state**: $\hat{x}_k = A\hat{x}_{k-1} + \Pi_k K_k \mathcal{I}_k + w_k$.

**end for**

---

$w_k \sim \mathcal{N}(0, Q_k - \Pi_k M_k \Pi_k^\top)$ is a Gaussian noise, uncorrelated with all other states, observations and noises in the system up to time $k$. Again, note that $\mathcal{I}_k$ is independent of the measurements up to time $k-1$, hence this choice makes $J_k$ independent of $\hat{X}_0^{k-1}$. These innovation-based corrections resemble the mechanism exploited in (8), hence we will refer to optimal filters of the form (24) as *perceptual* Kalman filters (PKF).

Now, by a straightforward substitution, (17) becomes

$$D_k = A_k D_{k-1} A_k^\top + Q_k + M_k - \Pi_k M_k - M_k \Pi_k^\top, \quad k = 0, \dots, T, \tag{25}$$

where we consider $Q_0 = P_0$, $M_0 = \Sigma_{\hat{x}_0^*}$ and $D_{-1} = 0$. As before, minimizing (5) boils down to minimizing $\sum_{k=0}^T \alpha_k \text{Tr}\{D_k\}$, and in order for (24) to be well-defined, we should enforce the constraints $Q_k - \Pi_k M_k \Pi_k^\top \succeq 0$, $k = 0, \dots, T$. For simplicity, we consider the time-invariant case where $A_k \equiv A$, so that the optimization objective becomes

$$\begin{cases} \min_{\{\Pi_k\}_{k=0}^T} & \sum_{k=0}^T \alpha_k \text{Tr}\{D_k\} \\ \text{s.t.} & D_k = \sum_{t=0}^k A^{k-t}\left[\mathcal{Q}_t - \Pi_t M_t - M_t \Pi_t^\top\right]\left(A^\top\right)^{k-t}, \quad k = 0, \dots, T, \\ & Q_k - \Pi_k M_k \Pi_k^\top \succeq 0, \quad k = 0, \dots, T, \end{cases} \tag{26}$$

where we denoted $\mathcal{Q}_k = Q_k + M_k$. Substituting $D_k$, we can rewrite the objective as

$$\sum_{k=0}^T \text{Tr}\left\{\sum_{t=k}^T \alpha_t A^{t-k}\left[\mathcal{Q}_k - 2\Pi_k M_k\right]\left(A^\top\right)^{t-k}\right\}. \tag{27}$$

As we can see, optimization over a particular coefficient $\Pi_k$ does not affect other summands of the external sum. Therefore, each $\Pi_k$ can be optimized separately. Minimizing the cost at the $k$-th step is equivalent to

$$\max_{\Pi_k} \text{Tr}\left\{\Pi_k M_k \sum_{t=k}^T \alpha_t (A^{t-k})^\top A^{t-k}\right\} \quad \text{s.t.} \quad Q_k - \Pi_k M_k \Pi_k^\top \succeq 0. \tag{28}$$

Let us denote $B_k \triangleq \sum_{t=k}^T \alpha_t (A^{t-k})^\top A^{t-k} = \alpha_k I + A^\top B_{k+1} A$. As we now show, this optimization problem possesses a closed-form solution under a mild assumption (which is satisfied e.g. when $Q_k \succ 0$). The proof is given in Appendix F.

**Theorem 4.3.** *Assume that* $\text{Im}\{B_k M_k B_k\} \subseteq \text{Im}\{Q_k\}$ *for every* $k$. *Let* $M_B \triangleq B_k M_k B_k$ *and denote* $\Omega = \left\{\Pi_k : Q_k - \Pi_k M_k \Pi_k^\top \succeq 0\right\}$. *Then the optimal value in* (28) *is given by*

$$\max_{\Pi_k \in \Omega} \text{Tr}\{\Pi_k M_k B_k\} = \text{Tr}\left\{\left(M_B^{1/2} Q_k M_B^{1/2}\right)^{\frac{1}{2}}\right\}, \tag{29}$$

*and is achieved by the optimal coefficient (which is generally not unique)*

$$\Pi_k^* = Q_k M_B^{1/2} \left(M_B^{1/2} Q_k M_B^{1/2}\right)^{1/2\dagger} M_B^{\dagger 1/2} B_k. \tag{30}$$

For a closed form solution under the alternative assumption that $\text{Im}\{M_k\} \subseteq \text{Im}\{Q_k\}$, as well as a discussion of stationary filters, please see the Appendix. The Perceptual Kalman filter (PKF) obtained from Thm. 4.3 is summarized in Alg. 1.

# 5 Numerical demonstrations [1]

We now revisit our main question: *what is the cost of temporal consistency in online restoration?* In addition, as we have seen in Sec. 4.2, the relaxation $\Phi_k = 0$, yielding the Perceptual Kalman filters, reduces the complexity of computation, possibly at the cost of higher errors. It is natural, then, to ask what is the cost of this simplification. In the following experiments, we compare the performance of several filters; $\hat{x}^*_{\mathrm{kal}}$ and $\hat{x}_{\mathrm{tic}}$ correspond to the Kalman filter and the temporally-inconsistent filter (10) (which does not possess perfect-perceptual quality). The estimate $\hat{x}_{\mathrm{opt}}$

Table 1: List of demonstrated filters.

| | description | def. | perfect-perception | |
| | | | per-sample | temporal |
|---|---|---|---|---|
| $\hat{x}^*_{\mathrm{kal}}$ | Kalman filter | Alg.2 | ✗ | ✗ |
| $\hat{x}_{\mathrm{tic}}$ | Per-sample quality (no temporal) | (10) | ✓ | ✗ |
| $\hat{x}_{\mathrm{opt}}$ | Optimized filter | (18) | ✓ | ✓ |
| $\hat{x}_{\mathrm{direct}}$ | Directly optimized | App. C | ✓ | ✓ |
| $\hat{x}_{\mathrm{auc}}$ | PKF (total cost) | Alg. 1 | ✓ | ✓ |
| $\hat{x}_{\mathrm{minT}}$ | PKF (terminal cost) | Alg.1 | ✓ | ✓ |

is a perfect-perception filter obtained by numerically optimizing the coefficients in (18), where the cost is the MSE at termination time, $\mathcal{C}_{\mathrm{T}} = \mathbb{E}[\|\hat{x}_T - x_T\|^2]$. $\hat{x}_{\mathrm{direct}}$ is a perfect-perception filter obtained by the optimization approach discussed in Appendix C, where we consider again the terminal cost. The estimates $\hat{x}_{\mathrm{auc}}, \hat{x}_{\mathrm{minT}}$ correspond to PKF outputs (Alg. 1) minimizing the *total cost* (area under curve) $\mathcal{C}_{\mathrm{auc}} = \sum_{k=0}^{T} \mathbb{E}[\|\hat{x}_k - x_k\|^2]$ and the *terminal cost* $C_T$, respectively. The filters are summarized in Table 1. Full details and additional experimental results are given in App. I.

## 5.1 Harmonic oscillator

We start with a simple 2-D example. Specifically, we consider a harmonic oscillator, where the state $x_k \in \mathbb{R}^2$ corresponds to position and velocity, and the observation at time $t$ is a noisy versions of the position at time $t - \frac{1}{2}\Delta_t$, where $\Delta_t$ is the sampling period (see App. I for details). Figure 3 shows the MSE for $\hat{x}_{\mathrm{opt}}$ and the sub-optimal PKF outputs $\hat{x}_{\mathrm{auc}}, \hat{x}_{\mathrm{minT}}$. The estimates $\hat{x}^*_{\mathrm{kal}}$ and $\hat{x}_{\mathrm{tic}}$ achieve lower MSE than $\hat{x}_{\mathrm{opt}}$, however they do not possess perfect-perceptual quality. *The difference in MSE between the filters $\hat{x}_{\mathrm{opt}}$ and $\hat{x}_{\mathrm{tic}}$ is the cost of temporal consistency in online estimation for this setting.*

In Figure 3, we also observe that the PKF estimations are indeed not MSE optimal at time $T = 255$. However, their RMSE is only about

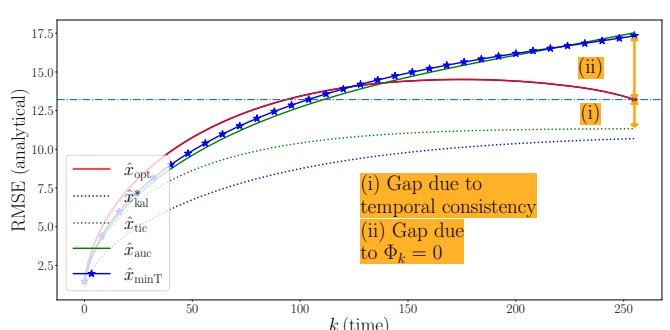

Figure 3: **MSE on Harmonic oscillator.** We observe $(i)$ the difference in distortion between the perfect-perceptual state $\hat{x}_{\mathrm{opt}}$, optimized according to (18), and $\hat{x}_{\mathrm{tic}}$. This additional cost is due to the perceptual constraint on the joint distribution. Also note the gap $(ii)$ between MSE of the optimized estimator and $\hat{x}_{\mathrm{minT}}$ which is due to the sub-optimal choice of coefficients, $\Phi_k = 0$.

30% higher than that of $\hat{x}_{\mathrm{opt}}$ and they have the advantage that they can be solved analytically and require computing only half of the coefficients ($\Pi_k$). The penalty related to this reduction may vary, depending on the exact setup. In Fig. 4 we demonstrate the relations between gaps due to temporal consistency (gap between $\hat{x}_{\mathrm{direct}}$ and $\hat{x}_{\mathrm{tic}}$) and due to the reduction $\Phi_k = 0$ (gap between $\hat{x}_{\mathrm{direct}}$ and $\hat{x}_{\mathrm{minT}}$), in various settings based on the Harmonic Oscillator example. On the *left* pane, dynamics are driven by $x_k = \rho A x_{k-1} + q_k$, where $A$ is a fixed marginally-stable matrix, and $\rho$ is a scalar controlling the strength of correlation between timesteps. On the *right* pane, $\rho = 1$ and observations are given only up to time $\tau$ ($y_k$ is only noise for $k \geq \tau$). We observe that for systems where timesteps are strongly correlated, or in the absence of current information, discarding $\upsilon_k$ leads

---

[1]Our code is publicly available at https://github.com/ML-group-il/perceptual-kalman-filters.

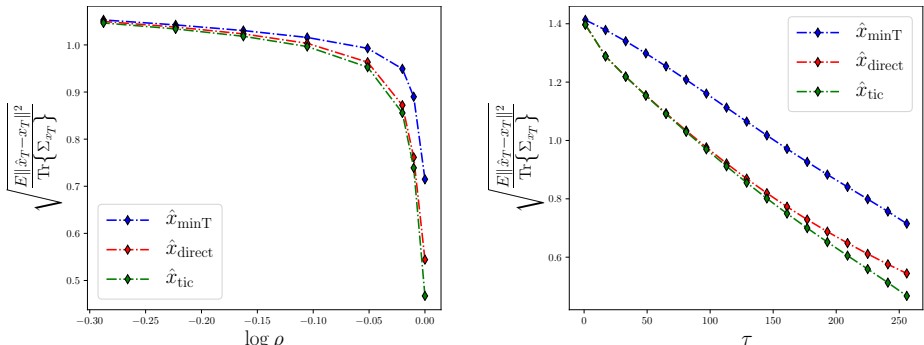

Figure 4: **Different dynamics yield different MSE gaps.** Here we use different dynamics, based on the Harmonic oscillator (matrices $A, C, Q, R$ are given in Sec. I.1) to demonstrate behavior of gaps due to temporal consistency (gap between $\hat{x}_{\text{direct}}$ and $\hat{x}_{\text{tic}}$) and due to the reduction $\Phi_k = 0$ (gap between $\hat{x}_{\text{direct}}$ and $\hat{x}_{\text{minT}}$). We present analytical errors at time $T = 255$, normalized by the ground-truth state $x_T$ variance. **(Left)** dynamics are driven by the stabilized matrix $\rho A$ for different scalar values of $\rho$. Observe that when timesteps are more correlated (higher $\rho$), both consistency and unutilized information play a major role, hence MSE gaps between filters grow. **(Right)** Here, $\rho = 1$, and observations are given only up to time $\tau$ ($y_k = $ noise, for $k \geq \tau$).

to significantly higher errors. Whenever states are less correlated and more observations are given, unutilized information can be ignored with lower cost.

## 5.2 Dynamic texture

We now illustrate the qualitative effects of perceptual estimation in a simplified video restoration setting. Specifically, we consider a video of a "dynamic texture" of waves in a lake. Such dynamic textures are accurately modeled by linear dynamics of a Gaussian latent representation [5], whose parameters we learn from a real video. Here, frames are generated from a latent $128$-dimensional state $x_k^{FA}$ which corresponds to their *Factor-Analysis* (FA) decomposition (see *e.g.* [2, Sec. 12.2.4] for more details). Thus, $512 \times 512 \times 3$ frames in the video domain are created through an affine transformation of $x_k^{FA}$. Linear observations $y_k \in \mathbb{R}^{32 \times 32}$ are given in the frame (pixel) domain, by $16\times$ downsampling the $Y$-channel of the generated ground-truth frames, and adding white Gaussian noise. All filtering is done in the latent domain, and then transformed to the pixel domain. MSE is also calculated in the FA domain. The exact settings can be found in App. I.

In the first experiment, measurements are supplied up to frame $k = 127$ and then stop (Fig. 5), letting the different filters predict the next, unobserved, frames of the sequence. We can see that until frame $k = 127$, all filters reconstruct the reference frames well. Starting from time $k = 128$, when measurements stop, the Kalman filter slowly fades into a static, blurry output which is the average frame value in this setting. This is a non-'realistic' video; Neither the individual frames nor the temporal (static) behavior are natural to the domain. Our perfect-perceptual filter, $\hat{x}_{\text{auc}}$, keeps generating a 'natural' video, both spatially and temporally. This makes its MSE grow faster[2].

We now perform a second experiment, where measurements are set to zero until frame $k = 512$. At times $k > 512$ they are given again by the noisy, downsampled frames as described above. In Fig. 6 we present the outcomes of the different filters. We first note that up to frame $k = 512$, there is no observed information, hence outputs are actually being generated according to priors. The Kalman filter outputs a static, average frame. The filter $\hat{x}_{\text{tic}}$ randomizes each frame independently, leading to unnatural random movement with flickering features. At frame $k = 513$, when observations become available, $\hat{x}_{\text{kal}}^*$ and $\hat{x}_{\text{tic}}$ get updated immediately, creating an inconsistent, non-smooth motion between frames 512 and 513. The PKF output $\hat{x}_{\text{auc}}$, on the other hand, maintains a smooth motion. Since the outputs of inconsistent filters rapidly becomes similar to the ground-truth, their errors drop. The perfect-perceptual filter, $\hat{x}_{\text{auc}}$, remains consistent with its previously generated frames and the natural dynamics of the model, hence its error decays more slowly.

---

[2]More visual details, including ground-truth clips and empirical error can be found in the Appendix. Full video clips for both experiments are supplied with the supplementary material.

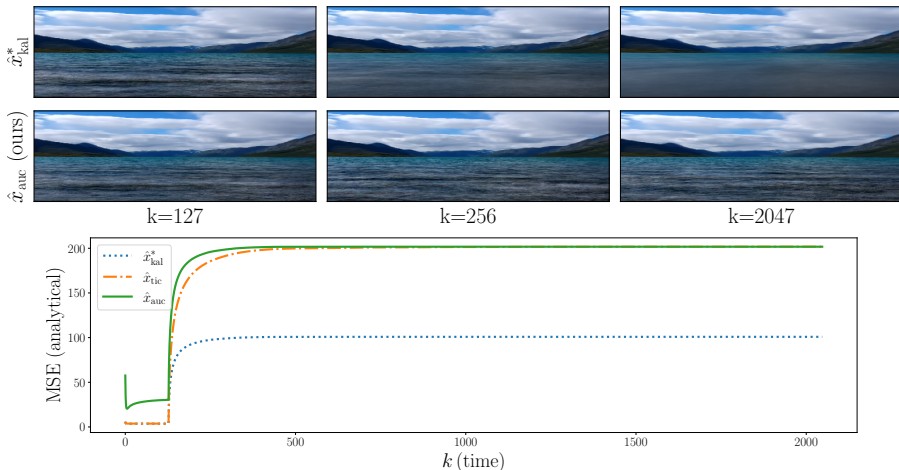

Figure 5: **Frame prediction on a dynamic texture domain.** In this experiment, measurements are supplied only up to frame $k = 127$ and the filter's task is to predict the unobserved future frames. Observe that $\hat{x}^*_{\mathrm{kal}}$ fades into a blurred average frame, while the perceptual filter $\hat{x}_{\mathrm{auc}}$ generates a natural video, both spatially and temporally. This makes its MSE grow faster.

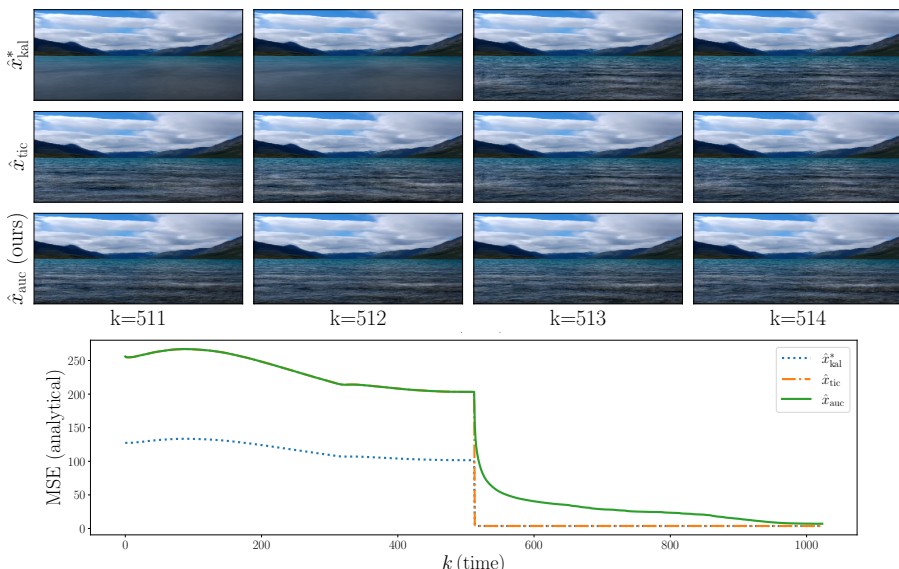

Figure 6: **Frame generation on a dynamic texture domain.** In the first half of the demo ($k \leq 512$), there are no observations, hence the reference signal is restored according to prior distribution. The filters with no perfect-perceptual quality constraint in the temporal domain generate non-realistic frames (Kalman filter output $\hat{x}^*_{\mathrm{kal}}$) or unnatural motion ($\hat{x}_{\mathrm{tic}}$). Perceptual filter $\hat{x}_{\mathrm{auc}}$ is constrained by previously generated frames and the natural dynamics of the domain, hence its MSE decays slower.

**Conclusion** We studied the problem of causal filtering of time sequences from corrupted or missing observations, where the the filter's output process is constrained to possess perfect (spatio-temporal) perceptual-quality in the sense of being distributed like the original signal. Our theoretical derivations focused on Gauss-Markov state-space processes. We introduced the novel concept of an unutilized information process and established a special class of perceptual filters, coined Perceptual Kalman Filters (PKF), that are based on the innovation process alone. We demonstrated the qualitative effects of perfect perceptual quality estimation on a video reconstruction problem. To the best of our knowledge, this is the first work addressing the distortion-perception tradeoff in online restoration settings. This work paves the way toward understanding perceptual online filtering, and the cost of temporal consistency in sequential estimation problems.

**Acknowledgements**   The work of RM was partially supported by the Israel Science Foundation grant 1693/22 and by the Skillman chair in biomedical sciences. The work of TM was partially supported by the Israel Science Foundation grant 2318/22 and by a gift from KLA. RM and TM were supported by the Ollendorff Minerva Center, ECE Faculty, Technion.

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

# Perceptual Kalman Filters: Online State Estimation under a Perfect Perceptual-Quality Constraint - Supplementary Material

In App. A we provide a detailed theoretical background on the Kalman Filter, its properties and recursive calculation. In App. B we prove that under our perfect perceptual filtering setting, there exists a *linear* optimal filter (Thm. 4.1). In App. C we discuss a direct, non-recursive method for optimizing perceptual filter coefficients. In App. D we present the derivation of the Lyapunov equation (17) for the error of perceptual filters. In App. E we derive the recursive expression for the filter given in (18). In App. F we find a closed-form solution for PKF coefficients by proving Theorem 4.3. In this appendix, we also give some brief overview on the extremal problem of finding a minimal distance between distributions. App.G contains a discussion about stationary perceptual Kalman filters in the steady-state regime. We summarize all definitions and notations in App. H. Finally, in App.I we give full details for all numerical demonstrations, and present additional empirical and visual results. More results are provided in the supplementary video.

## A    The Kalman Filter algorithm

In this Section we supply a detailed reminder of the Kalman filter Algorithm. The celebrated Kalman filter [11] assumes a state $x_k \in \mathbb{R}^{n_x}$, where dynamics are modeled as deterministic linear functions perturbed by a Gaussian noise, and observations $y_k \in \mathbb{R}^{n_y}$ are linear functions of $x_k$ with an additive noise

$$
\begin{aligned}
x_k &= A_k x_{k-1} + q_k, & q_k &\sim \mathcal{N}(0, Q_k), & k &= 1, ..., T, & (31) \\
y_k &= C_k x_k + r_k, & r_k &\sim \mathcal{N}(0, R_k), & k &= 0, ..., T. & (32)
\end{aligned}
$$

The noise terms $q_k$ and $r_k$ are independent white Gaussian processes with zero mean and covariances $Q_k, R_k$, respectively. $x_0$ is assumed to have a zero-mean Gaussian distribution with covariance $P_0$, independent of $q_1, r_0$. For convenience, we will sometimes refer to $P_0$ as $Q_0$. The matrices $A_k, C_k, Q_k. R_k$ and $P_0$ are system parameters with appropriate dimensions, and assumed to be known. Considering the MSE distortion, we denote

$$
\hat{x}_{k|s} \triangleq \operatorname*{argmin}_{\hat{x}} \mathbb{E}\left[\|x_k - \hat{x}\|^2 | y_0, \ldots, y_s\right], \tag{33}
$$

namely the optimal MSE estimator of the state at time $k$, given measurements up to time $s$. Under the assumptions mentioned above, Kalman filters produce the mean state estimate $\hat{x}_{k|k}$, an MSE-optimal estimator of $x_k$ given the observations up to time $k$. The *Kalman optimal state* $\hat{x}_k^* \equiv \hat{x}_{k|k}$ is given by the recurrence

$$
\hat{x}_k^* = A_k \hat{x}_{k-1}^* + K_k \mathcal{I}_k, \tag{34}
$$

where $K_k$ is the *optimal Kalman gain* [11], given explicitly in Algorithm 2.

The vector $\mathcal{I}_k$ is the *innovation* process,

$$
\mathcal{I}_k = y_k - C_k \hat{x}_{k|k-1}, \tag{35}
$$

describing the contribution of the new observation $y_k$ over the optimal prediction based on previous observations. Since we are in the Linear-Gaussian setup, we have that the innovation state $\mathcal{I}_k$ is orthogonal to the measurements $y_0, \ldots, y_{k-1}$, guaranteeing the MSE optimality of the estimation. The calculation of Kalman state is summarized in Algorithm 2.

---
**Algorithm 2** Kalman Filter
---
**initialize:** $\hat{x}_0^* = K_0 y_0 = P_0 C_0^\top \Sigma_{Y_0}^{-1} y_0, \quad P_{0|0} = P_0 - P_0 C_0^\top \Sigma_{Y_0}^{-1} C_0 P_0, \mathcal{I}_0 = y_0, S_0 = C_0 P_0 C_0^\top + R_0.$

**for** $k = 1$ to $T$ **do**

    calculate prior: $\hat{x}_{k|k-1} = A_k \hat{x}_{k-1|k-1}, \quad P_{k|k-1} = A_k P_{k-1|k-1} A_k^\top + Q_k$

    calculate Innovation: $\mathcal{I}_k = y_k - C_k \hat{x}_{k|k-1}, \quad S_k = C_k P_{k|k-1} C_k^\top + R_k$

    Kalman gain: $K_k = P_{k|k-1} C_k^\top S_k^{-1}$

    update (posterior): $\hat{x}_k^* = \hat{x}_{k|k} = \hat{x}_{k|k-1} + K_k \mathcal{I}_k, \quad P_{k|k} = (I - K_k C_k) P_{k|k-1}$

**end for**
---

# B Optimality of linear filters (proof of Thm. 4.1)

In this section we show that under a family of optimality criteria (5) and perfect-perceptual quality and causality constraints (6-7), linear filters of the form (14) are optimal. We start with the following.

**Theorem B.1.** *Let $Y_0^T = (y_0, \ldots, y_T)$ be the set of measurements (4), and let $(J_0^T, Y_0^T)$ be a joint distribution s.t. $J_k$ is independent of $y_{k+n}$ given $Y_0^k$ for all $k \in [0, T]$ and $n \geq 1$. Then,*

$$\mathbb{E}\left[J_k \mathcal{I}_{k+n}^\top\right] = 0. \tag{36}$$

*Proof.* Denote $\hat{J}_k = \mathbb{E}\left[J_k | \mathcal{I}_0^k\right]$. We can write the measurements as a linear function of the innovations, $y_k = \sum_{t=0}^k H_{k,t} \mathcal{I}_t$. We have

$$\hat{y}_{k+n}^{k+n-1} \triangleq \mathbb{E}\left[y_{k+n} | Y_0^{k+n-1}\right] = \mathbb{E}\left[y_{k+n} | \mathcal{I}_0^{k+n-1}\right] = \sum_{t=0}^{k+n-1} H_{k+n,t} \mathcal{I}_t, \tag{37}$$

and

$$\hat{y}_{k+n}^k \triangleq \mathbb{E}\left[y_{k+n} | Y_0^k\right] = \mathbb{E}\left[y_{k+n} | \mathcal{I}_0^k\right] = \sum_{t=0}^k H_{k+n,t} \mathcal{I}_t = \mathbb{E}\left[\hat{y}_{k+n}^{k+n-1} | \mathcal{I}_0^k\right]. \tag{38}$$

For any $k$ and $n = 1$, $\mathcal{I}_{k+1} = y_{k+1} - \hat{y}_{k+1}^k$, and therefore

$$\mathbb{E}\left[J_k \mathcal{I}_{k+1}^\top\right] = \mathbb{E}\left[\mathbb{E}\left[J_k \left[y_{k+1} - \hat{y}_{k+1}^k\right]^\top | \mathcal{I}_0^k\right]\right] = \mathbb{E}\left[\hat{J}_k \left[\hat{y}_{k+1}^k\right]^\top\right] - \mathbb{E}\left[J_k | \mathcal{I}_0^k\right]\left[\hat{y}_{k+1}^k\right]^\top\right] = 0. \tag{39}$$

This is due to the facts that $J_k$ and $y_{k+1}$ are independent given the condition, and that $\hat{y}_{k+1}^k$ is a deterministic function of $\mathcal{I}_0^k$.

Now, assume we know that $\mathbb{E}\left[J_k \mathcal{I}_t^\top\right] = 0$ for $k + 1 \leq t \leq k + n - 1$. We can write

$$\begin{aligned}
\mathbb{E}\left[J_k \mathcal{I}_{k+n}^\top\right] &= \mathbb{E}\left[\mathbb{E}\left[J_k \left[y_{k+n} - \hat{y}_{k+n}^{k+n-1}\right]^\top | \mathcal{I}_0^k\right]\right] \\
&= \mathbb{E}\left[\hat{J}_k \left[\hat{y}_{k+n}^k\right]^\top\right] - \mathbb{E}\left[J_k \sum_{t=0}^{k+n-1} \mathcal{I}_t^\top H_{k+n,t}^\top | \mathcal{I}_0^k\right] \\
&= \mathbb{E}\left[\hat{J}_k \left[\hat{y}_{k+n}^k\right]^\top\right] - \mathbb{E}\left[J_k \sum_{t=0}^k \mathcal{I}_t^\top H_{k+n,t}^\top | \mathcal{I}_0^k\right] - \mathbb{E}\left[J_k \sum_{t=k+1}^{k+n-1} \mathcal{I}_t^\top H_{k+n,t}^\top | \mathcal{I}_0^k\right] \\
&= \mathbb{E}\left[\hat{J}_k \left[\hat{y}_{k+n}^k\right]^\top - \hat{J}_k \left[\hat{y}_{k+n}^k\right]^\top - \sum_{t=k+1}^{k+n-1} \mathbb{E}\left[J_k \mathcal{I}_t^\top | \mathcal{I}_0^k\right] H_{k+n,t}^\top\right] \\
&= -\sum_{t=k+1}^{k+n-1} \mathbb{E}\left[J_k \mathcal{I}_t^\top\right] H_{k+n,t}^\top \\
&= 0. \tag{40}
\end{aligned}$$

$\square$

We now show that for every filter which is feasible under (6) and (7), one can find a linear filter, jointly Gaussian with the measurement set, attaining the same cost objective.

**Theorem B.2.** *Let $Y_0^T = (y_0, \ldots, y_T)$ be the set of measurements (4), and let $\mathcal{J}_0^T = (\mathcal{J}_0, \ldots, \mathcal{J}_T)$ be jointly distributed with $Y_0^T$ such that:*

(i) $\mathcal{J}_0^T \sim \mathcal{N}\left(0, \mathrm{diag}\{P_0, Q_1, \ldots, Q_T\}\right)$.

(ii) $\mathcal{J}_k$ *is independent of $y_{k+n}$ given $Y_0^k$ for all $k \in [0, T]$ and $n \geq 1$.*

(iii) $\sum_{k=0}^T \alpha_k \mathbb{E}\left[\|x_k - \chi_k\|^2\right] = \mathcal{C}$*, where $\chi_k$ is the process given by $\chi_k = A_k \chi_{k-1} + \mathcal{J}_k$ with $\chi_0 = \mathcal{J}_0$.*

*Then, there exists a joint Gaussian distribution $(J_0^T, Y_0^T)$ in which* (i) *and* (ii) *hold, and the estimator given by*

$$\hat{x}_k = A_k \hat{x}_{k-1} + J_k, \quad \hat{x}_0 = J_0 \tag{41}$$

*achieves the same cost* (iii), *namely* $\sum_{k=0}^{T} \alpha_k \mathbb{E}\left[\|x_k - \hat{x}_k\|^2\right] = C.$

*Furthermore, we can write*

$$J_k = \pi_k \mathcal{I}_k + \phi_k \upsilon_k + w_k, \tag{42}$$

*where*

$$\upsilon_k = \mathcal{I}_0^{k-1} - \mathbb{E}\left[\mathcal{I}_0^{k-1}|J_0^{k-1}\right] \tag{43}$$

*and $w_k$ is a white Gaussian noise, independent of $Y_0^T$ and $J_0^{k-1}$.*

*Proof.* Let $(J_0^T, Y_0^T)$ be the Gaussian distribution defined by the moments of $(\mathcal{J}_0^T, Y_0^T)$ up to second order. We observe that from Theorem B.1 above, $J_k$ is independent of all future innovations $\mathcal{I}_{k+n}$, namely it is based only on measurements up to time $k$. Using the notions of Theorem B.1's proof,

$$
\begin{aligned}
\mathbb{E}\left[(J_k - \hat{J}_k)(y_{k+n} - \hat{y}_{k+n}^k)^\top | Y_0^k\right] &= \mathbb{E}\left[(J_k - \hat{J}_k) \sum_{t=k+1}^{k+n} \mathcal{I}_t^\top H_{k+n,t}^\top | \mathcal{I}_0^k\right] \\
&= \sum_{t=k+1}^{k+n} \left[\mathbb{E}\left[J_k \mathcal{I}_t^\top | \mathcal{I}_0^k\right] - \hat{J}_k \mathbb{E}\left[\mathcal{I}_t^\top | \mathcal{I}_0^k\right]\right] H_{k+n,t}^\top \\
&= \sum_{t=k+1}^{k+n} \mathbb{E}\left[\mathbb{E}\left[J_k \mathcal{I}_t^\top | \mathcal{I}_t, \mathcal{I}_0^k\right] | \mathcal{I}_0^k\right] H_{k+n,t}^\top \\
&= \sum_{t=k+1}^{k+n} \mathbb{E}\left[\mathbb{E}\left[J_k | \mathcal{I}_t, \mathcal{I}_0^k\right] \mathcal{I}_t^\top | \mathcal{I}_0^k\right] H_{k+n,t}^\top \\
&= \sum_{t=k+1}^{k+n} \mathbb{E}\left[\mathbb{E}\left[J_k | \mathcal{I}_0^k\right] \mathcal{I}_t^\top | \mathcal{I}_0^k\right] H_{k+n,t}^\top \\
&= \sum_{t=k+1}^{k+n} \mathbb{E}\left[J_k | \mathcal{I}_0^k\right] \mathbb{E}\left[\mathcal{I}_t^\top | \mathcal{I}_0^k\right] H_{k+n,t}^\top \\
&= 0. \tag{44}
\end{aligned}
$$

This means that $J_k$ and $y_{k+n}$ are independent given $Y_0^k$, which proves (ii).

From (17) we see that the cost functional depends only on the second order statistics of $(\mathcal{J}_0^T, \mathcal{I}_0^T)$ which are identical to those of $(J_0^T, \mathcal{I}_0^T)$, hence (iii) holds:

$$\sum_{k=0}^{T} \alpha_k \mathbb{E}\left[\|x_k - \hat{x}_k\|^2\right] = \sum_{k=0}^{T} \alpha_k \mathbb{E}\left[\|x_k - \chi_k\|^2\right] = \mathcal{C}. \tag{45}$$

To prove (42), we now write

$$J_k = \varepsilon_k + w_k, \tag{46}$$

where $\varepsilon_k = \mathbb{E}\left[J_k | Y_0^T, J_0^{k-1}\right]$, and $w_k = J_k - \mathbb{E}\left[J_k | Y_0^T, J_0^{k-1}\right]$ is independent of $Y_0^T$ and $J_0^{k-1}$. Now, since both $J_k$ and $J_0^{k-1}$ are independent of $\mathcal{I}_{k+1}^T$,

$$\varepsilon_k = \mathbb{E}\left[J_k | Y_0^T, J_0^{k-1}\right] = \mathbb{E}\left[J_k | \mathcal{I}_0^k, J_0^{k-1}\right] = \sum_{t=0}^{k} \phi_{k,t} \mathcal{I}_t + \sum_{t=0}^{k-1} \psi_{k,t} J_t. \tag{47}$$

$J_k$ is independent of $J_0^{k-1}$, thus

$$\mathbb{E}\left[J_k | J_0^{k-1}\right] = \mathbb{E}\left[\mathbb{E}\left[J_k | \mathcal{I}_0^T, J_0^{k-1}\right] | J_0^{k-1}\right] = 0. \tag{48}$$

Conditioning both sides of (47) on $J_0^{k-1}$ and taking expectations,

$$0 = \sum_{t=0}^{k} \phi_{k,t} \mathbb{E}\left[\mathcal{I}_t | J_0^{k-1}\right] + \sum_{t=0}^{k-1} \psi_{k,t} J_t. \tag{49}$$

Note that $\mathbb{E}\left[\mathcal{I}_k | J_0^{k-1}\right] = 0$, which together with (49) implies

$$\varepsilon_k = \phi_{k,k} \mathcal{I}_k + \sum_{t=0}^{k-1} \phi_{k,t} \left[\mathcal{I}_t - \mathbb{E}\left[\mathcal{I}_t | J_0^{k-1}\right]\right] = \pi_k \mathcal{I}_k + \phi_k \upsilon_k. \tag{50}$$

Now, all we have left to show is that $w_k$ is a white sequence. Since $w_{k+n}$ ($n \geq 1$) is independent of $J_0^k$ and $\mathcal{I}_0^T$ (which also constitute $\upsilon_k$), it is easy to obtain

$$\mathbb{E}\left[w_{k+n} w_k^\top\right] = \mathbb{E}\left[w_{k+n} \left[J_k - \pi_k \mathcal{I}_k - \phi_k \upsilon_k\right]^\top\right] = 0. \tag{51}$$

$\square$

**Corollary B.3.** *Given a cost objective of the form $\mathcal{C} = \sum_{k=0}^{T} \alpha_k \mathbb{E}\left[\|x_k - \hat{x}_k\|^2\right]$, there exists a linear filter of the form*

$$J_k = \pi_k \mathcal{I}_k + \phi_k \upsilon_k + w_k, \tag{52}$$

*such that*

$$\hat{x}_0 = J_0 \tag{53}$$
$$\hat{x}_k = A_k \hat{x}_{k-1} + J_k, \ k = 1, \ldots, T \tag{54}$$

*is an optimal estimator under the perfect perceptual quality and causality constraints (6-7).*

*Proof.* Under the perfect perceptual quality constraint, an estimate sequence $\chi_k$ must satisfy that

$$\mathcal{J}_k = \chi_k - A_k \chi_{k-1} \tag{55}$$

is a white Gaussian process with covariances $Q_k$. If, in addition, $\chi_k$ satisfies the causality condition (6), so does $\mathcal{J}_k$. We conclude from Theorem B.2 that there exists a causal linear filter $J_k$ that achieves the same expected objective $\mathcal{C}$ as $\chi_k$.

Now, note again that from (17), for perfect-perceptual quality causal filters, the objective $\mathcal{C}$ is a continuous function of the covariance matrix

$$\mathbb{E}\left[\mathcal{J}_0^T \left(\mathcal{I}_0^T\right)^\top\right] = \begin{bmatrix} \text{diag}\{P_0, Q_1, \ldots, Q_T\} & L \\ L^\top & \text{diag}\{S_0, S_1, \ldots, S_T\} \end{bmatrix} \succeq 0, \tag{56}$$

where, due to the causality demand, $L$ is a quasi lower triangular matrix. The set of such feasible matrices is non-empty, closed (since it is the intersection of the closed cone of PSD matrices with a finite set of hyperplanes) and bounded. Hence, $\mathcal{C}$ attains a minimal value on some joint distribution $p_{\mathcal{J}_0^T, Y_0^T}$, which can be chosen to be joint-Gaussian as we have seen.

$\square$

## C   A Direct optimization approach to perfect-perceptual quality filtering

For the sake of completeness, we now discuss a method for optimizing non-recursive perfect-perceptual quality filter coefficients. This approach leads to convex programs. However, as we will see next, it might become impractical for large configurations.

Let $J = J_0^T \sim \mathcal{N}(0, Q)$, where $Q = \text{diag}\left\{\{Q_k\}_{k=0}^T\right\}$, be a causal function of the measurements, $J = \Phi\mathcal{I} + W$, where $\mathcal{I} = \mathcal{I}_0^T$ is the innovation process with covariance $S = \text{diag}\{S_k\}$ and $W$ is an independent noise. Now, $\hat{X} = \hat{X}_0^T = A_J J$ is the filter's output, where

$$A_J = \begin{bmatrix} I & 0 & \dots & 0 \\ A_1 & I & \dots & 0 \\ \vdots & \vdots & \ddots & \vdots \\ \prod_{k=0}^{T-1} A_{T-k} & \prod_{k=1}^{T-1} A_{T-k} & \dots & I \end{bmatrix}. \tag{57}$$

Recall $X^*$ is the Kalman filter output given by $X^* = A_J K\mathcal{I}$, where $K = \text{diag}\{K_k\}$. Let $\mathscr{W} = \text{diag}\{\alpha_k\} \otimes I_{n_x}$ be a weighting matrix. The objective (5) is now given by

$$\begin{aligned} \mathcal{C}(\hat{X}) &= \mathbb{E}\left[(\hat{X} - X^*)^\top \mathscr{W}(\hat{X} - X^*)\right] \\ &= \text{Tr}\left\{\mathscr{W}\mathbb{E}\left[\hat{X}\hat{X}^T\right] + \mathscr{W}\mathbb{E}\left[X^* X^{*\top}\right] - 2\mathscr{W}\mathbb{E}\left[\hat{X}X^{*\top}\right]\right\}. \end{aligned} \tag{58}$$

Hence, we have to maximize

$$\begin{aligned} \mathcal{C}(\Phi) &= 2\text{Tr}\left\{\mathscr{W}\mathbb{E}\left[\hat{X}X^{*\top}\right]\right\} \\ &= 2\text{Tr}\left\{\mathscr{W}A_J \Phi S K^\top A_J^\top\right\} \\ &= 2\text{Tr}\left\{(\Phi S)K^\top A_J^\top \mathscr{W}A_J\right\} \\ &= 2\text{Tr}\left\{\Phi S K^\top B\right\}, \end{aligned} \tag{59}$$

where $B \triangleq A_J^\top \mathscr{W}A_J$. This is subject to the perfect perceptual-quality constraint

$$Q - \Phi S \Phi^\top \succeq 0, \text{ or equivalently } \begin{bmatrix} Q & \Phi S \\ S\Phi^\top & S \end{bmatrix} \succeq 0, \tag{60}$$

where $\Phi$ is a lower quasi-triangular matrix (causality constraint)

$$\Phi = \begin{bmatrix} \Phi_{0,0} & 0 & \dots & 0 \\ \Phi_{1,0} & \Phi_{1,1} & \dots & 0 \\ \vdots & \vdots & \ddots & \vdots \\ \Phi_{T,0} & \Phi_{T,1} & \dots & \Phi_{T,T} \end{bmatrix}. \tag{61}$$

Again, under this formulation,

$$J = \Phi\mathcal{I} + W, \tag{62}$$

where $W \sim \mathcal{N}\left(0, Q - \Phi S \Phi^\top\right)$ is a Gaussian noise independent of $\mathcal{I}$. Note that $W_0^T$ might not be a white sequence in this case, since its covariance might not be a block-diagonal matrix. As a result, the noise sequence has to be sampled dependently. Also note that this problem possesses the same memory complexity as (14). To conclude, this method leads to convex, but large optimization programs, and is impractical for high dimensional settings or long temporal sequences.

## D  Derivation of eq. (17)

Recall $\hat{x}_k^*$ is the optimal Kalman state at time $k$, achieving MSE given by

$$d_k^* = \mathbb{E}\left[\|\hat{x}_k^* - x_k\|^2\right] = \mathrm{Tr}\left\{P_{k|k}\right\}. \tag{63}$$

$P_{k|k}$ is the error covariance, given explicitly in Algorithm 2. By the orthogonality principle, for any estimator $\hat{x}_k$ based on the measurements $y_0, \ldots, y_k$ we have

$$\mathbb{E}\left[\|x_k - \hat{x}_k\|^2\right] = \mathbb{E}\left[\|x_k - \hat{x}_k^*\|^2\right] + \mathbb{E}\left[\|\hat{x}_k^* - \hat{x}_k\|^2\right] = d_k^* + \mathbb{E}\left[\|\hat{x}_k - \hat{x}_k^*\|^2\right]. \tag{64}$$

Now, consider an estimator $\hat{x}_k$ of the form (12), and recall

$$D_k \triangleq \mathbb{E}\left[\left[\hat{x}_k^* - \hat{x}_k\right]\left[\hat{x}_k^* - \hat{x}_k\right]^\top\right]. \tag{65}$$

Since we choose $J_k \sim \mathcal{N}(0, Q_k)$ to be independent of $\hat{x}_{k-1}$ and $\mathcal{I}_k$ is indepenedent of $\hat{x}_{k-1}$ and $\hat{x}_{k-1}^*$, we write

$$
\begin{aligned}
D_k &= \mathbb{E}\left[\hat{x}_k^* - \hat{x}_k\right]\left[\hat{x}_k^* - \hat{x}_k\right]^\top \\
&= \mathbb{E}\left[A_k\hat{x}_{k-1} - A_k\hat{x}_{k-1}^* + J_k - K_k\mathcal{I}_k\right]\left[A_k\hat{x}_{k-1} - A_k\hat{x}_{k-1}^* + J_k - K_k\mathcal{I}_k\right]^\top \\
&= A_k\mathbb{E}\left[\left[\hat{x}_{k-1}^* - \hat{x}_{k-1}\right]\left[\hat{x}_{k-1}^* - \hat{x}_{k-1}\right]^\top\right]A_k^\top + \mathbb{E}\left[J_k J_K^\top\right] + K_k\mathbb{E}\left[\mathcal{I}_k\mathcal{I}_k^\top\right]K_k^\top \\
&\quad - \mathbb{E}\left[J_k\left[A_k\hat{x}_{k-1}^* + K_k\mathcal{I}_k\right]^\top\right] - \mathbb{E}\left[\left[A_k\hat{x}_{k-1}^* + K_k\mathcal{I}_k\right]J_k^\top\right] \\
&= A_k D_{k-1} A_k^\top + K_k S_k K_k^\top + Q_k \\
&\quad - \mathbb{E}\left[J_k\mathcal{I}_k^\top\right]K_k^\top - K_k\mathbb{E}\left[\mathcal{I}_k J_k^\top\right] - A_k\mathbb{E}\left[\hat{x}_{k-1}^* J_k^\top\right] - \mathbb{E}\left[J_k\hat{x}_{k-1}^{*\top}\right]A_k^\top. \tag{66}
\end{aligned}
$$

# E Derivation of recursive perfect-perceptual quality filters

We now derive the recursive expression (21)-(22) for the filter given in (18),

$$\hat{x}_k = A_k \hat{x}_{k-1} + J_k, \tag{67}$$

$$J_k = \Phi_k A_k \Upsilon_k + \Pi_k K_k \mathcal{I}_k + w_k, \ w_k \sim \mathcal{N}\left(0, \Sigma_{w_k}\right), \tag{68}$$

defined by the coefficients $\{\Pi_k, \Phi_t\}_{t=0}^T$ fulfilling the constraints (20). Recall

$$\Upsilon_k \triangleq \hat{x}_{k-1}^* - \mathbb{E}\left[\hat{x}_{k-1}^* | \hat{x}_0, \dots, \hat{x}_{k-1}\right] = \hat{x}_{k-1}^* - \mathbb{E}\left[\hat{x}_{k-1}^* | J_0, \dots, J_{k-1}\right] \tag{69}$$

where $\hat{x}_k^*$ is the Kalman state. $J_0^{k-1}, \Upsilon_k, \mathcal{I}_k, w_k$ are jointly-Gaussian and independent, and we have

$$\mathbb{E}\left[J_n J_k^\top\right] = Q_k \delta_{n=k}, \tag{70}$$

$$\mathbb{E}\left[\mathcal{I}_k J_k^\top\right] = S_k K_k^\top \Pi_k^\top, \tag{71}$$

$$\mathbb{E}\left[\Upsilon_k J_k^\top\right] = \Sigma_{\Upsilon_k} A_k^\top \Phi_k^\top. \tag{72}$$

We can write

$$\Upsilon_{k+1} - A_k \Upsilon_k = \hat{x}_k^* - A_k \hat{x}_{k-1}^* - \left[\mathbb{E}\left[\hat{x}_k^* | J_0^k\right] - A_k \mathbb{E}\left[\hat{x}_{k-1}^* | J_0^{k-1}\right]\right]$$

$$= K_k \mathcal{I}_k - K_k \mathbb{E}\left[\mathcal{I}_k | J_0^k\right] - A_k \left[\mathbb{E}\left[\hat{x}_{k-1}^* | J_0^k\right] - \mathbb{E}\left[\hat{x}_{k-1}^* | J_0^{k-1}\right]\right] \tag{73}$$

Since $J_0^k$ is an independent sequence, and since $\mathcal{I}_k$ depends only on $J_k$,

$$K_k \mathbb{E}\left[\mathcal{I}_k | J_0^k\right] = K_k \mathbb{E}\left[\mathcal{I}_k | J_k\right] = K_k S_k K_k^\top \Pi_k^\top Q_k^\dagger J_k. \tag{74}$$

We also have that $\Upsilon_k, J_k$ are independent of $J_0^{k-1}$, implying

$$\mathbb{E}\left[\hat{x}_{k-1}^* | J_0^k\right] - \mathbb{E}\left[\hat{x}_{k-1}^* | J_0^{k-1}\right] = \mathbb{E}\left[\hat{x}_{k-1}^* - \mathbb{E}\left[\hat{x}_{k-1}^* | J_0^{k-1}\right] | J_0^k\right]$$

$$= \mathbb{E}\left[\Upsilon_k | J_0^k\right] = \mathbb{E}\left[\Upsilon_k | J_k\right]$$

$$= \Sigma_{\Upsilon_k} A_k^\top \Phi_k^\top Q_k^\dagger J_k. \tag{75}$$

Hence,

$$\Upsilon_{k+1} = A_k \Upsilon_k + K_k \mathcal{I}_k - \Psi_k Q_k^\dagger J_k, \tag{76}$$

where we denote

$$\Psi_k \triangleq M_k \Pi_k^\top + A_k \Sigma_{\Upsilon_k} A_k^\top \Phi_k^\top. \tag{77}$$

The covariance is then given by the recursive form

$$\Sigma_{\Upsilon_{k+1}} = A_k \Sigma_{\Upsilon_k} A_k^\top + M_k + \Psi_k Q_k^\dagger \Psi_k^\top$$

$$- A_k \Sigma_{\Upsilon_k} A_k^\top \Phi_k^\top Q_k^\dagger \Psi_k^\top - K_k S_k K_k^\top \Pi_k^\top Q_k^\dagger \Psi_k^\top \tag{78}$$

$$- \left[A_k \Sigma_{\Upsilon_k} A_k^\top \Phi_k^\top Q_k^\dagger \Psi_k^\top\right]^\top - \left[K_k S_k K_k^\top \Pi_k^\top Q_k^\dagger \Psi_k^\top\right]^\top \tag{79}$$

$$= A_k \Sigma_{\Upsilon_k} A_k^\top + M_k - \Psi_k Q_k^\dagger \Psi_k^\top. \tag{80}$$

At time $k = 0$ we have $\Upsilon_0 = 0$ and $\Sigma_{\Upsilon_0} = 0$.

*Remark* E.1 (The non-reduced case). For the full, non-reduced linear filter (14)- (15) , we have the following similar formula

$$\upsilon_k = \begin{bmatrix} \mathcal{I}_{k-1} \\ \upsilon_{k-1} \end{bmatrix} - \begin{bmatrix} S_{k-1} & 0 \\ 0 & \Sigma_{\upsilon_{k-1}} \end{bmatrix} \begin{bmatrix} \pi_{k-1}^\top \\ \phi_{k-1}^\top \end{bmatrix} Q_{k-1}^\dagger J_{k-1} \tag{81}$$

and

$$\Sigma_{\upsilon_k} = \begin{bmatrix} S_{k-1} & 0 \\ 0 & \Sigma_{\upsilon_{k-1}} \end{bmatrix} - \begin{bmatrix} S_{k-1} & 0 \\ 0 & \Sigma_{\upsilon_{k-1}} \end{bmatrix} \begin{bmatrix} \pi_{k-1}^\top \\ \phi_{k-1}^\top \end{bmatrix} Q_{k-1}^\dagger \begin{bmatrix} \pi_{k-1}^\top \\ \phi_{k-1}^\top \end{bmatrix}^\top \begin{bmatrix} S_{k-1} & 0 \\ 0 & \Sigma_{\upsilon_{k-1}} \end{bmatrix}. \tag{82}$$

Notice, however, that the dimension of $\upsilon_k$ grows with time $k$.

# F   A Generalized extremal problem with semidefinite constraints (proof of Thm. 4.3)

In this section we prove Theorem 4.3. We start with a brief overview of the extremal problem of finding a minimal distance between distributions, and of general semi-definite programs.

To prove the Theorem we observe that (28), is a generalization of the extremal problem, and suggest a non-trivial dual form where, under our assumptions, strong duality holds.

## F.1   Minimal distance between distributions

Consider two Gaussian distributions on $\mathbb{R}^n$ with zero means and PSD covariance matrices $\Sigma_1, \Sigma_2$ respectively. We consider the problem of constructing a Gaussian vector $[X, Y]$ minimizing $\mathbb{E}\|X - Y\|^2$ while inducing the given marginal distributions, $X \sim \mathcal{N}(0, \Sigma_1), Y \sim \mathcal{N}(0, \Sigma_2)$. This problem is equivalent to the following maximization of correlation [13]

$$\mathrm{Tr}\{2\Pi\} \to \max_\Pi, \quad \text{s.t.} \, \Sigma = \begin{bmatrix} \Sigma_1 & \Pi \\ \Pi^\top & \Sigma_2 \end{bmatrix} \succeq 0. \tag{83}$$

We have the following results of Olkin and Pukelsheim [13].

**Lemma F.1.** *[13, Lemma 1]. Let $\Sigma_2^g$ be any generalized inverse of $\Sigma_2$. Then $\Sigma \succeq 0$ iff*

$$\Sigma_2 \Sigma_2^g \Pi^\top = \Pi^\top \text{ and } \Sigma_1 - \Pi \Sigma_2^g \Pi^\top \succeq 0. \tag{84}$$

**Theorem F.2.** *[13, Thm. 4]. If $\mathrm{Im}\{\Sigma_2\} \subseteq \mathrm{Im}\{\Sigma_1\}$, then an optimal solution to (83) is given by*

$$\max_\Pi \mathrm{Tr}\{2\Pi\} = 2\mathrm{Tr}\left\{\left(\Sigma_2^{1/2} \Sigma_1 \Sigma_2^{1/2}\right)^{1/2}\right\}, \tag{85}$$

*achieved by the argument*

$$\Pi^* = \Sigma_1 \Sigma_2^{1/2} \left[\left(\Sigma_2^{1/2} \Sigma_1 \Sigma_2^{1/2}\right)^{1/2}\right]^g \Sigma_2^{1/2}. \tag{86}$$

*In the case where $\mathrm{Im}\{\Sigma_2\} = \mathrm{Im}\{\Sigma_1\}$, $\Pi^*$ is a unique optimal argument.*

Under the setting discussed in Sec. 2, Theorem F.2 implies that in the more general case where $\Sigma_x \succeq 0$, the MSE-optimal perfect perceptual-quality estimator (2) is obtained by

$$\hat{x} = \mathscr{T}^* x^* + w, \quad \mathscr{T}^* \triangleq \Sigma_x \Sigma_{x^*}^{\frac{1}{2}} (\Sigma_{x^*}^{\frac{1}{2}} \Sigma_x \Sigma_{x^*}^{\frac{1}{2}})^{\frac{1}{2}\dagger} \Sigma_{x^*}^{\frac{1}{2}\dagger}. \tag{87}$$

Here again, $w$ is a zero-mean Gaussian noise with covariance $\Sigma_w = \Sigma_x - \mathscr{T}^* \Sigma_{x^*} \mathscr{T}^{*\top}$, independent of $y$ and $x$, and $\Sigma_{x^*}^\dagger$ is the Moore-Penrose inverse of $\Sigma_{x^*}$.

## F.2   SDP Setting and duality - background

*Semi-definite programming* (SDP) [9, 17] is an optimization problem in $X \in \mathbb{R}^{n \times n}$ of the form

$$C \bullet X \to \max_X \tag{88}$$

$$\text{s.t. } A_i \bullet X = b_i, i = 1, \ldots, m, \tag{89}$$

$$X \succeq 0. \tag{90}$$

Here, $C, A_i$ are real symmetric matrices of appropriate dimensions, and $A \bullet X = \mathrm{Tr}\{A^\top X\}$ is the Frobenius product. SDPs yield the Lagrangian

$$L(X, \lambda, \nu) = \nu^\top b + \left(C - \sum_{i=0}^m \nu_i A_i\right) \bullet X + \lambda \rho_{min}(X)$$

$$= \nu^\top b + \left(C - \sum_{i=0}^m \nu_i A_i\right) \bullet X + \min_{Y \succeq 0, \mathrm{Tr} Y = \lambda} Y \bullet X, \tag{91}$$

where $\lambda \geq 0$ and $\rho_{min}$ is the minimal eigenvalue. The Dual problem (DSP) is given by

$$\nu^\top b \to \min_\nu, \quad \text{s.t.} \, C - \sum_{i=0}^m \nu_i A_i \preceq 0. \tag{92}$$

In this case, strong duality exists iff the SDP is strictly feasible, *i.e.* it has a feasible solution interior to the feasible set, $X \succ 0$. This condition is sometimes referred to as the *Slater condition*.

## F.3 A generalized extremal problem with strong duality

Recall $Q_k$, $M_k$, $B_k$ are real, symmetric positive semidefinite $n_x \times n_x$ matrices, and the optimization problem (28),

$$\text{Tr}\left\{\tilde{\Pi}_k M_k B_k\right\} = \text{Tr}\left\{\tilde{\Pi}_k M_k M_k^\dagger M_k B_k\right\} \to \max_{\tilde{\Pi}_k}, \text{ s.t. } Q_k - \tilde{\Pi}_k M_k \tilde{\Pi}_k^\top \succeq 0. \qquad (93)$$

Since (93) involves a single time step, we will omit the index $k$.

We consider $\Pi = \tilde{\Pi} M$, hence $\Pi^\top = M M^\dagger \Pi^\top$, and since $M = M M^\dagger M$ we rewrite (93) as

$$\text{Tr}\left\{\Pi B\right\} = \text{Tr}\left\{B\Pi\right\} \to \max_{\Pi}, \text{ s.t., } Q - \Pi M^\dagger \Pi^\top \succeq 0, \Pi^\top = M M^\dagger \Pi^\top. \qquad (94)$$

By Lemma F.1, the constraints in (94) are equivalent to

$$X \triangleq \begin{bmatrix} Q & \Pi \\ \Pi^\top & M \end{bmatrix} \succeq 0. \qquad (95)$$

This can be formulated as the semi-definite program,

$$C \bullet X \to \max_{X}, \text{ s.t. } \begin{cases} A_{ij}^Q \bullet X &= Q_{ij} \\ A_{ij}^M \bullet X &= M_{ij} \end{cases}, 0 \le i \le j \le n-1, X \succeq 0, \qquad (96)$$

where $C = \frac{1}{2}\begin{bmatrix} 0 & B \\ B & 0 \end{bmatrix}$, and $A_{ij}^Q = \begin{bmatrix} \Lambda_{ij} & 0 \\ 0 & 0 \end{bmatrix}$, $A_{ij}^M = \begin{bmatrix} 0 & 0 \\ 0 & \Lambda_{ij} \end{bmatrix}$, $\Lambda_{ij} = \frac{1}{2}(e_{ij} + e_{ji})$.

Note that when $B$ is a scalar matrix, (94) is similar to the problem studied in Olkin and Pukelsheim [13]. Their approach was later extended by Shapiro [15] to general linear objectives, where the Slater condition holds.

### F.3.1 Strong duality

The SDP (96) yields the standard dual formulation

$$Q \bullet \nu_Q + M \bullet \nu_M \to \min_{\nu_Q, \nu_M}, \text{ s.t. } \begin{bmatrix} \nu_Q & -\frac{1}{2}B \\ -\frac{1}{2}B & \nu_M \end{bmatrix} \succeq 0, \nu_Q, \nu_M \in \mathbb{R}^{n_x \times n_x}. \qquad (97)$$

This should give us a hint about the optimal solution to (94). Pay attention, however, that according to the theory, strong duality in (97) is guaranteed only if $Q, M \succ 0$, which might not be the case (see *e.g.* [15]). To get a tight bound for the general case $Q, M \succeq 0$, we now provide an alternative form of duality to (94).

The following is an adaptation of techniques used in Olkin and Pukelsheim [13]. We start with the following Lemma.

**Lemma F.3.** *Let $\Pi$ be a feasible solution to (94), $R, G \in \mathbb{R}^{n_x \times n_x}$ are general matrices. Then,*

$$\text{Tr}\left\{QRR^\top + BMBGG^\top\right\} \ge 2\text{Tr}\left\{\Pi BGR^\top\right\}. \qquad (98)$$

*Proof.* From the non-negativity of $X$ in (95) we have

$$\begin{bmatrix} R^\top, -G^\top B \end{bmatrix}\begin{bmatrix} Q & \Pi \\ \Pi^\top & M \end{bmatrix}\begin{bmatrix} R \\ -BG \end{bmatrix} = $$
$$R^\top Q R + G^\top B M B G - R^\top \Pi B G - G^\top B \Pi^\top R \succeq 0. \qquad (99)$$

The trace is nonnegative, hence we have the desired result. $\qquad \square$

*Remark* F.4. Similarly, we can obtain

$$\text{Tr}\left\{QBRR^\top B + MGG^\top\right\} \ge 2\text{Tr}\left\{B\Pi GR^\top\right\}. \qquad (100)$$

Now, we suggest an alternative to (DSP) (97), where strong duality will hold.

**Theorem F.5.** *[Strong duality]. Let*

$$\Omega = \left\{ \Pi \in \mathbb{R}^{n_x \times n_x} : Q - \Pi M^\dagger \Pi^\top \succeq 0, \Pi^\top = MM^\dagger \Pi^\top \right\}, \tag{101}$$

$$\mathcal{S} = \left\{ (S, S^-) : S, S^- \succeq 0, SS^- S = S, S^- SS^- = S^-, BM = SS^- BM \right\}, \tag{102}$$

*and denote $M_B \triangleq BMB$. Assume $\mathrm{Im}\,\{M_B\} \subseteq \mathrm{Im}\,\{Q\}$. Then,*

$$\min_{(S,S^-)\in\mathcal{S}} \left\{ Q \bullet S + M \bullet (BS^- B) \right\} = \max_{\Pi\in\Omega} \mathrm{Tr}\,\{2\Pi B\}$$
$$= 2\mathrm{Tr}\left\{ \left( M_B^{1/2} Q M_B^{1/2} \right)^{1/2} \right\}. \tag{103}$$

*The extreme value is obtained for*

$$S^* = M_B^{1/2} \left( M_B^{1/2} Q M_B^{1/2} \right)^{1/2\dagger} M_B^{1/2}, \tag{104}$$

$$S^{-*} = M_B^{1/2\dagger} \left( M_B^{1/2} Q M_B^{1/2} \right)^{1/2} M_B^{1/2\dagger}, \tag{105}$$

$$\Pi^* = QS^* M_B^\dagger BM = Q M_B^{1/2} \left( M_B^{1/2} Q M_B^{1/2} \right)^{1/2\dagger} M_B^{1/2\dagger} BM. \tag{106}$$

*Optimal solution $\Pi^*$ is generally not unique.*

To prove strong duality, we will use the following lemmas.

**Lemma F.6.** *Assume PSD matrices $Q, M_B$ such that $\mathrm{Im}\,\{M_B\} \subseteq \mathrm{Im}\,\{Q\}$, then $\mathrm{Im}\,\{M_B\} = \mathrm{Im}\left\{ M_B^{1/2} Q M_B^{1/2} \right\}$.*

*Proof.* Recall $M_B, M_B^{1/2} Q M_B^{1/2}$ are real symmetric matrices.

Let $v \in \mathrm{Ker}\{M_B^{1/2} Q M_B^{1/2}\}$, we have $\|Q^{1/2} M_B^{1/2} v\| = 0$ hence $M_B^{1/2} v \in \mathrm{Ker}\{Q^{1/2}\} \subseteq \mathrm{Ker}\{M_B^{1/2}\}$, which yields $M_B v = 0$, implying $\mathrm{Ker}\{M_B^{1/2} Q M_B^{1/2}\} \subseteq \mathrm{Ker}\{M_B\}$. Opposite relation is trivial.

We have

$$\mathrm{Im}\,\{M_B\} = \mathrm{Ker}\{M_B\}^\perp = \mathrm{Ker}\{M_B^{1/2} Q M_B^{1/2}\}^\perp = \mathrm{Im}\left\{ M_B^{1/2} Q M_B^{1/2} \right\}. \tag{107}$$

□

**Lemma F.7.** $\mathrm{Im}\,\{BM\} \subseteq \mathrm{Im}\,\{BMB\}$.

*Proof.* Let $v \in \mathrm{Ker}\{BMB\}$, then $\|M^{1/2} Bv\| = 0$ and $Bv \in \mathrm{Ker}\{M^{1/2}\} = Ker\{M\}$. Hence $\mathrm{Ker}\{BMB\} \subseteq \mathrm{Ker}\{MB\}$. We have

$$\mathrm{Im}\,\{BM\} = \mathrm{Ker}\{MB\}^\perp \subseteq \mathrm{Ker}\{BMB\}^\perp = \mathrm{Im}\,\{BMB\}. \tag{108}$$

□

We are now ready to prove Theorem F.5.

*Proof.* [Theorem F.5]. Let $\Pi \in \Omega$, then $X \succeq 0$ in (95). For any $(S, S^-) \in \mathcal{S}$ we can choose $R = S^{1/2}, G = S^- R$. From the result of Lemma F.3 it follows that

$$Q \bullet S + M \bullet (BS^- B) = \mathrm{Tr}\left\{ QRR^\top + BMBGG^\top \right\}$$
$$\geq 2\mathrm{Tr}\left\{ \Pi BGR^\top \right\} = 2\mathrm{Tr}\left\{ \Pi BS^- S \right\} = 2\mathrm{Tr}\left\{ \Pi B \right\}. \tag{109}$$

The last equality holds since $BM = SS^- BM$, and $\mathrm{Im}\left\{ \Pi^T \right\} \subseteq \mathrm{Im}\,\{M\}$.

We now prove that $\Pi^* \in \Omega$.

$$
\begin{aligned}
&Q - \Pi^* M^\dagger \Pi^{*\top} \\
&= Q - QM_B^{1/2}\left(M_B^{1/2}QM_B^{1/2}\right)^{1/2\dagger} M_B^{\dagger 1/2} BMM^\dagger MBM_B^{\dagger 1/2}\left(M_B^{1/2}QM_B^{1/2}\right)^{1/2\dagger} M_B^{1/2}Q \\
&= Q - QM_B^{1/2}\left(M_B^{1/2}QM_B^{1/2}\right)^{\dagger} M_B^{1/2}Q \\
&= Q^{1/2}\left[I - Q^{1/2}M_B^{1/2}\left(M_B^{1/2}QM_B^{1/2}\right)^{\dagger} M_B^{1/2}Q^{1/2}\right]Q^{1/2} \\
&= Q^{1/2}\left[I - Q^{1/2}M_B^{1/2}\left(M_B^{1/2}QM_B^{1/2}\right)^{\dagger} M_B^{1/2}Q^{1/2}\right]^2 Q^{1/2} \succeq 0.
\end{aligned}
$$
$$(110)$$

The last equality holds since it is easy to see that $\left[I - Q^{1/2}M_B^{1/2}\left(M_B^{1/2}QM_B^{1/2}\right)^{\dagger} M_B^{1/2}Q^{1/2}\right]$ is a symmetric (orthogonal) projection.

We further prove that $S^*, S^{-*} \in \mathcal{S}$. It is easy to show that $S^*, S^{-*}$ are symmetric generalized inverses, reflexive to each other ($S^{-*}$ is in fact the Moore-Penrose inverse of $S^*$):

$$
S^* S^{-*} = M_B^{1/2}\left(M_B^{1/2}QM_B^{1/2}\right)^{1/2\dagger} M_B^{1/2} M_B^{1/2\dagger}\left(M_B^{1/2}QM_B^{1/2}\right)^{1/2} M_B^{1/2\dagger} \tag{111}
$$

$$
= M_B^{1/2}\left(M_B^{1/2}QM_B^{1/2}\right)^{1/2\dagger}\left(M_B^{1/2}QM_B^{1/2}\right)^{1/2} M_B^{1/2\dagger} \tag{112}
$$

$$
= M_B^{1/2} M_B^{1/2\dagger} = M_B^{1/2\dagger} M_B^{1/2} \tag{113}
$$

$$
= S^{-*} S^*. \tag{114}
$$

The equalities hold since by Lemma F.6,

$$
\mathrm{Im}\left\{M_B^{1/2}\right\} = Im\{M_B\} = \mathrm{Im}\left\{M_B^{1/2}QM_B^{1/2}\right\} = \mathrm{Im}\left\{\left(M_B^{1/2}QM_B^{1/2}\right)^{1/2}\right\}, \tag{115}
$$

and since for a PSD matrix $R$, $RR^\dagger = R^\dagger R$ is an orthogonal projection onto its image. Using Lemma F.7 we have

$$
S^* S^{-*} BM = M_B^{1/2\dagger} M_B^{1/2} BM = BM. \tag{116}
$$

It is now easy to verify that

$$
Q \bullet S^* + M \bullet \left(BS^{-*}B\right) = 2\mathrm{Tr}\left\{\Pi^* B\right\} = 2\mathrm{Tr}\left\{\left(M_B^{1/2}QM_B^{1/2}\right)^{1/2}\right\}, \tag{117}
$$

which completes the proof. $\qquad\square$

**Corollary F.8.** *Under the assumption,* $\mathrm{Im}\left\{M_B\right\} \subseteq \mathrm{Im}\left\{Q_k\right\}$, *the optimal gain in* (28) *is given by*

$$
\Pi_k^* = Q_k M_B^{1/2}\left(M_B^{1/2}Q_k M_B^{1/2}\right)^{1/2\dagger} M_B^{\dagger 1/2} B_k. \tag{118}
$$

*Remark F.9.* Under the alternative assumption, $\mathrm{Im}\left\{M_k\right\} \subseteq \mathrm{Im}\left\{Q_k\right\}$, the optimal gain in (28) is given by

$$
\Pi_k^* = Q_k \tilde{B} M_b^{1/2}\left(M_b^{1/2}Q_b M_b^{1/2}\right)^{1/2\dagger} M_b^{\dagger 1/2} \tilde{B}, \tag{119}
$$

where $\tilde{B} = B_k^{1/2}$, $Q_b = \tilde{B}Q_k\tilde{B}$, $M_b = \tilde{B}M_k\tilde{B}$.

*Proof.* Recall our goal in (28) is to maximize $\mathrm{Tr}\left\{\Pi MB\right\} = \mathrm{Tr}\left\{\tilde{B}\Pi M\tilde{B}\right\}$ under the condition $Q - \Pi M\Pi^\top \succeq 0$ (we omit the index $k$). This is equivalent to minimizing $\mathbb{E}\left[\|\tilde{B}X - \tilde{B}Y\|^2\right]$ w.r.t $\Pi$, where $(X, Y) \sim \mathcal{N}(0, \Sigma)$ and $\Sigma = \begin{bmatrix} Q & \Pi M \\ M\Pi^\top & M \end{bmatrix} \succeq 0$.

In this case, $(\tilde{B}X, \tilde{B}Y) \sim \mathcal{N}(0, \Sigma_b)$ where $\Sigma_b = \begin{bmatrix} \tilde{B}Q\tilde{B} & \tilde{B}\Pi M\tilde{B} \\ \tilde{B}M\Pi^\top\tilde{B} & \tilde{B}M\tilde{B} \end{bmatrix}$. According to Thm. F.2, under the assumption $\mathrm{Im}\left\{\tilde{B}M\tilde{B}\right\} \subseteq \mathrm{Im}\left\{\tilde{B}Q\tilde{B}\right\}$, the minimal distance is achieved when

$$\tilde{B}\Pi M\tilde{B} = \tilde{B}Q\tilde{B}M_b^{\frac{1}{2}}\left(M_b^{\frac{1}{2}}Q_b M_b^{\frac{1}{2}}\right)^{\frac{1}{2}\dagger}M_b^{\frac{1}{2}}. \tag{120}$$

Note that $\mathrm{Im}\{M\} \subseteq \mathrm{Im}\{Q\}$ implies $\mathrm{Im}\left\{\tilde{B}M\tilde{B}\right\} \subseteq \mathrm{Im}\left\{\tilde{B}Q\tilde{B}\right\}$, and it is straightforward to verify that $Q - \Pi^* M \Pi^{*\top} \succeq 0$. $\qquad\square$

# G Stationary settings

A note is in place regarding the stationary perceptual Kalman filter. In the Kalman steady-state regime, where dynamics (31) -(32) are time-invariant and $T \to \infty$, the matrices $K$ and $S$ in Algorithm 2 are determined by the covariance matrix $P$,

$$K = PC^\top(CPC^\top + R)^{-1}, \quad S = CPC^\top + R. \tag{121}$$

Here, $C$ stands for the time-invariant observation matrix ($y_k = Cx_k + r_k$) and $P$ is a solution to the Discrete-Time Algebraic Riccati equation (DARE)

$$P = APA^\top - APC^\top(CPC^\top + R)^{-1}CPA^\top + Q. \tag{122}$$

Similarly, under the steady-state regime, (26) becomes

$$\begin{cases} \mathrm{Tr}\,\{D\} & \to \min_\Pi \\ \text{s.t.} & D = ADA^\top + Q + M - \Pi M - M\Pi^\top, M = KSK^\top, Q - \Pi M\Pi^\top \succeq 0 \end{cases} \tag{123}$$

where $D$ obeys an (Algebraic) Lyapunov equation. If $A$ is stable,

$$D(\Pi) = \sum_{k=0}^\infty A^k(Q + M - \Pi M - M\Pi^\top)\left(A^k\right)^\top. \tag{124}$$

Hence, stationary perceptual filter is of the form

$$\hat{x}_k = A\hat{x}_{k-1} + J_k, \tag{125}$$
$$J_k = \Pi K \mathcal{I}_k + w_k, \tag{126}$$
$$w_k \sim \mathcal{N}\left(0, Q - \Pi M\Pi^\top\right), \tag{127}$$

and in order to find optimal gain $\Pi$, minimizing $\mathrm{Tr}\,\{D(\Pi)\}$, we have to solve

$$\max_\Pi \mathrm{Tr}\,\{\Pi MB\} \text{ s.t. } Q - \Pi M\Pi^\top \succeq 0, \tag{128}$$

where we define $B \triangleq \sum_{k=0}^\infty \left(A^k\right)^\top A^k$, and the solution (under the assumption $\mathrm{Im}\,\{BMB\} \subseteq \mathrm{Im}\,\{Q\}$) is given again by (30).

## H   List of notations

We summarize our notations in the following Table.

Table 2: Definitions and Notations

| Notation | Description | Definition | Dimensions |
|---|---|---|---|
| $n_x$ | state dimension | | |
| $n_y$ | measurement dimension | | |
| $A_k$ | system dynamics | | $n_x \times n_x$ |
| $C_k$ | measurement function | | $n_y \times n_x$ |
| $Q_k, R_k$ | noise covariances | | $n_x \times n_x, n_y \times n_y$ |
| $x_k$ | system state (ground-truth) | | $n_x$ |
| $y_k$ | measurements | | $n_y$ |
| $\hat{x}_k$ | state estimator | | $n_x$ |
| $\hat{x}_k^*$ | optimal Kalman state | see Algorithm 2 | $n_x$ |
| $\hat{x}_{k|s}$ | best MSE state esimators, up to time $s$ | | $n_x$ |
| $\mathcal{I}_k$ | innovation process | see Algorithm 2 | $n_y$ |
| $S_k$ | innovation covariance | see Algorithm 2 | $n_y \times n_y$ |
| $K_k$ | Kalman gain | see Algorithm 2 | $n_x \times n_y$ |
| $\Pi_k$ | innovation perceptual gain | | $n_x \times n_x$ |
| $M_k$ | Kalman update covariance | $M_k = K_k S_k K_k^\top$ | $n_x \times n_x$ |
| $\upsilon_k$ | unutilized information process | see (15) | $k n_y$ |
| $\Upsilon_k$ | unutilized information process (recursive) | see (19) | $n_x$ |
| $\Sigma_{\Upsilon_k}$ | unutilized information covariance | | $n_x \times n_x$ |
| $\Phi_k$ | unutilized information perceptual gain | | $n_x \times n_x$ |
| $B_k$ | weight matrix | $B_k = \sum_{t=k}^{T} \alpha_t (A^{t-k})^\top A^{t-k}$ | $n_x \times n_x$ |
| $D_k$ | deviation from MMSE | $D_k = \mathbb{E}\left[\hat{x}_k^* - \hat{x}_k\right]\left[\hat{x}_k^* - \hat{x}_k\right]^\top$ | $n_x \times n_x$ |
| $T$ | Termination time (horizon) | | |
| $\mathcal{C}(\hat{X}_0^T)$ | minimization objective | $\mathcal{C} = \sum_{k=0}^{T} \alpha_k \mathbb{E}\left[\|x_k - \hat{x}_k\|^2\right]$ | |

# I Numerical demonstrations

In this section we provide full details for the experimental settings of Sec. 5, with additional numerical and visual results. In the following, we compare the performance of several filters; $\hat{x}_{\text{kal}}^*$ and $\hat{x}_{\text{tic}}$ correspond to the Kalman filter and the temporally-inconsistent filter (10) (which does not possess perfect-perceptual quality). The estimate $\hat{x}_{\text{opt}}$ is generated by a perfect-perception filter obtained by numerically optimizing the coefficients in (18), where the cost is the MSE at termination time, *i.e.* the *terminal cost*

$$\mathcal{C}_{\text{T}} = \mathbb{E}\left[\|\hat{x}_T - x_T\|^2\right]. \tag{129}$$

The estimates $\hat{x}_{\text{auc}}, \hat{x}_{\text{minT}}$ correspond to PKF outputs (Alg. 1) minimizing the *total cost* (area under curve)

$$\mathcal{C}_{\text{auc}} = \sum_{k=0}^{T} \mathbb{E}\left[\|\hat{x}_k - x_k\|^2\right], \tag{130}$$

and the *terminal cost*, respectively. Finally, $\hat{x}_{\text{stat.}}$ is the stationary PKF, discussed in App. G. The filters are summarized in Table 3.

Table 3: List of demonstrated filters.

| | description | definition | perfect-perception | |
| | | | per-sample | temporal |
|---|---|---|---|---|
| $\hat{x}_{\text{kal}}^*$ | Kalman filter | Algorithm 2 | ✗ | ✗ |
| $\hat{x}_{\text{tic}}$ | Per-sample perceptual quality (temporally-inconsistent) | (10) | ✓ | ✗ |
| $\hat{x}_{\text{opt}}$ | Optimized perfect-perceptual quality filter | (18) | ✓ | ✓ |
| $\hat{x}_{\text{auc}}$ | PKF with total cost | Algorithm 1 | ✓ | ✓ |
| $\hat{x}_{\text{minT}}$ | PKF with terminal cost | Algorithm 1 | ✓ | ✓ |
| $\hat{x}_{\text{stat.}}$ | Stationary PKF | (125) | ✗ | ✗ |

## I.1 Example: Harmonic oscillator

We start with a simple 2-D example, where we demonstrate the differences in MSE distortion between the optimized perfect-perceptual quality filter $\hat{x}_{\text{opt}}$, the temporally inconsistent filter $\hat{x}_{\text{tic}}$ and the efficient sub-optimal (perceptual) PKF. Consider the harmonic oscillator, where the entries of the state $x_k \in \mathbb{R}^2$ correspond to position and velocity, and evolve as

$$x_{k+1} = Ax_k + q_k, \quad q_k \sim \mathcal{N}(0, I) \tag{131}$$

with

$$A = I + \begin{bmatrix} 0 & 1 \\ -2 & 0 \end{bmatrix} \times \Delta_t, \tag{132}$$

where $\Delta_t = 5 \times 10^{-3}$ is the sampling interval. Assume we have access to noisy and delayed scalar observations of the position (corresponding to time $t - \frac{1}{2}\Delta_t$) so that $y_k = \begin{bmatrix} 1 & -\frac{1}{2} \end{bmatrix} x_k + r_k$, where $r_k \sim \mathcal{N}(0, 1)$ and $x_0 \sim \mathcal{N}(0, 0.8I)$.

We numerically optimize the coefficients $\{\Pi_k, \Phi_k\}_{k=0}^{T}$ in (18), to minimize the terminal error (129) ($\text{Tr}\{D_T\}$ in (23)) at time $T = 255$ under the constraints (20). Figure 7 shows the MSE distortion for the optimized perfect-perception filter $\hat{x}_{\text{opt}}$ defined by (18) and $\{\Pi_k, \Phi_k\}_{k=0}^{T}$, and the sub-optimal PKF outputs $\hat{x}_{\text{auc}}, \hat{x}_{\text{minT}}$, minimizing the total cost (130) and the terminal cost (129) (see Table 3). We observe that PKF estimations are indeed not MSE optimal at time $T$, However, their RMSE at time $T$ is only $\sim 30\%$ higher than that of $\hat{x}_{\text{opt}}$ and they have the advantage that they can be solved analytically and require computing only half of the coefficients ($\Pi_k$).

The estimates $\hat{x}_{\text{kal}}^*$ and $\hat{x}_{\text{tic}}$ achieve lower MSE than $\hat{x}_{\text{opt}}$, however they do not possess perfect-perceptual quality. We can see the difference in MSE distortion between the filters $\hat{x}_{\text{opt}}$ and $\hat{x}_{\text{tic}}$, with and without perception constraint in the temporal domain. *This is the cost of temporal consistency in online estimation for this setting.*

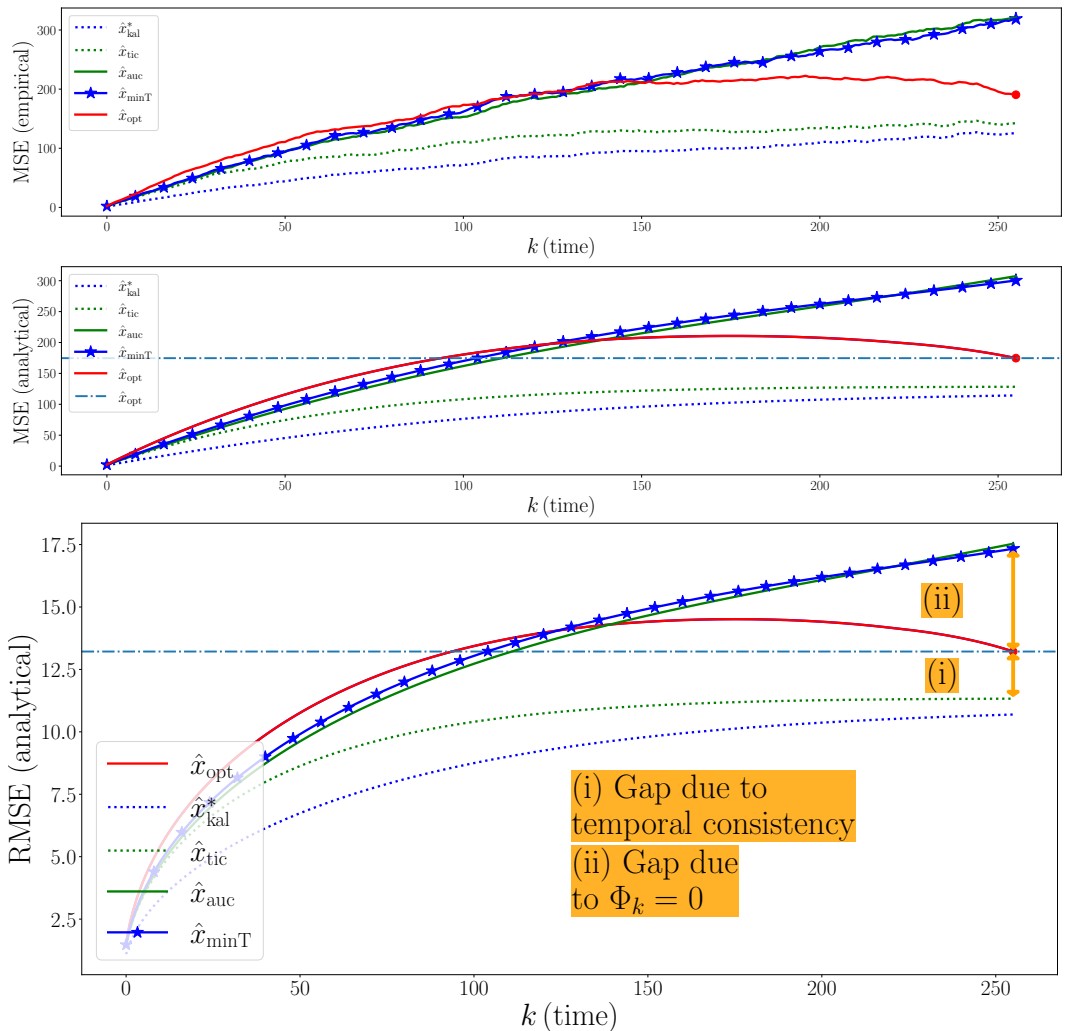

Figure 7: **MSE distortion on Harmonic oscillator.** $\hat{x}_{\mathrm{opt}}$ is a numerically optimized perfect-perceptual quality filter's output (minimizing error at time $T = 255$, dashed horizontal line). $\hat{x}_{\mathrm{auc}}, \hat{x}_{\mathrm{minT}}$ are PKF outputs minimizing different objectives. Observe that PKF estimations are not MSE optimal, but require less computations. $\hat{x}^*_{\mathrm{kal}}$ and $\hat{x}_{\mathrm{tic}}$ are not perfect-perceptual quality filters. **(top)** Empirical error, over $N = 1024$ sampled trajectories. **(bottom)** Analytical error. The difference in distortion between the perfect-perceptual state $\hat{x}_{\mathrm{opt}}$, optimized according to (18), and $\hat{x}_{\mathrm{tic}}$ is due to the perceptual constraint on the joint distribution. This is the cost of temporal consistency in online estimation for this setting. The gap between the MSE of the optimized estimator and $\hat{x}_{\mathrm{minT}}$ is due to the sub-optimal choice of coefficients, $\varPhi_k = 0$.

## I.2  Example: Two coupled inverted pendulums

Next, we demonstrate the quantitative behavior of perceptual Kalman filters, by comparing the MSE between the PKF outputs when minimizing different cost functions, and between non-perceptual filters outputs. More specifically, this experiment demonstrates:

1. How minimizing different objectives in Algorithm 1 leads to different filters.
2. The cost of perfect-perceptual quality filtering, given by Algorithm 1, over optimal filters.

We consider a higher-dimensional, well-studied example of two coupled inverted pendulums, mounted on carts [7, 6]. The cart positions, pendulum deviations, and their velocities (Fig. 8), are given by the discretized stable closed-loop system with perturbation

$$x_{k+1} = Ax_k + q_k, \quad q_k \sim \mathcal{N}(0, Q), \tag{133}$$

where $x_k \in \mathbb{R}^8$. The initial state is distributed as

$$x_0 \sim \mathcal{N}(0, P_0). \tag{134}$$

The system matrices are given by

$$A = I + A_{cl} \cdot \Delta_t, \tag{135}$$

where $\Delta_t = 5 \times 10^{-4}$ is the sampling interval and

$$A_{cl} = \begin{bmatrix} A_1 + BK_1 & F \\ F & A_2 + BK_2 \end{bmatrix}, \tag{136}$$

$$A_1 = A_2 = \begin{bmatrix} 0 & 1 & 0 & 0 \\ 2.9156 & 0 & -0.0005 & 0 \\ 0 & 0 & 0 & 1 \\ -1.6663 & 0 & 0.0002 & 0 \end{bmatrix}, \quad B = \begin{bmatrix} 0 \\ -0.0042 \\ 0 \\ 0.0167 \end{bmatrix}. \tag{137}$$

The *coupling* is given by

$$F = \begin{bmatrix} 0 & 0 & 0 & 0 \\ 0.0011 & 0 & 0,0.0005 & 0 \\ 0 & 0 & 0 & 0 \\ -0.0003 & 0 & -0.0002 & 0 \end{bmatrix}, \tag{138}$$

and stabilizing state-feedback controllers (each acts on a single cart) are

$$K_1 = [11396.0 \quad 7196.2 \quad 573.96 \quad 1199.0], \quad K_2 = [29241 \quad 18135 \quad 2875.3 \quad 3693.9]. \tag{139}$$

The partial measurements are given by $y_k = Cx_k + r_k$, where $r_k \sim \mathcal{N}(0, R)$, with coefficients

$$C = \begin{bmatrix} \bar{C}_1 & 0 \\ 0 & \bar{C}_2 \end{bmatrix}, \quad \bar{C}_1 = \bar{C}_2 = \begin{bmatrix} 1 & 0 & 0 & 0 \\ 0 & 0 & 1 & 0 \end{bmatrix}. \tag{140}$$

Namely, we observe only position and angle for each cart/pendulum, while velocities are not being measured.

The perturbation covariances are given by

$$P_0 = \begin{bmatrix} \bar{P}_0 & 0 \\ 0 & \bar{P}_0 \end{bmatrix}, \quad Q = \begin{bmatrix} \bar{Q} & 0 \\ 0 & \bar{Q} \end{bmatrix}, \quad R = \begin{bmatrix} \bar{R} & \frac{1}{8}\bar{R} \\ \frac{1}{8}\bar{R} & \bar{R} \end{bmatrix}, \tag{141}$$

where

$$\bar{P}_0 = \begin{bmatrix} 0.154 & 0.142 & -0.143 & 0.093 \\ 0.142 & 0.144 & -0.124 & 0.058 \\ -0.143 & -0.124 & 0.167 & -0.148 \\ 0.093 & 0.058 & -0.148 & 0.192 \end{bmatrix} \cdot 5 \times 10^{-4}, \tag{142}$$

$$\bar{Q} = 10^{-2} \cdot \begin{bmatrix} 0.642 & -0.136 & 0.78 & 0.262 \\ -0.136 & 0.894 & -0.248 & 0.074 \\ 0.78 & -0.248 & 1.284 & -0.314 \\ 0.262 & 0.074 & -0.314 & 1.766 \end{bmatrix} \times \Delta_t, \quad \bar{R} = 10^{-2} \cdot \begin{bmatrix} 0.375 & -0.33 \\ -0.33 & 0.771 \end{bmatrix} \times \Delta_t. \tag{143}$$

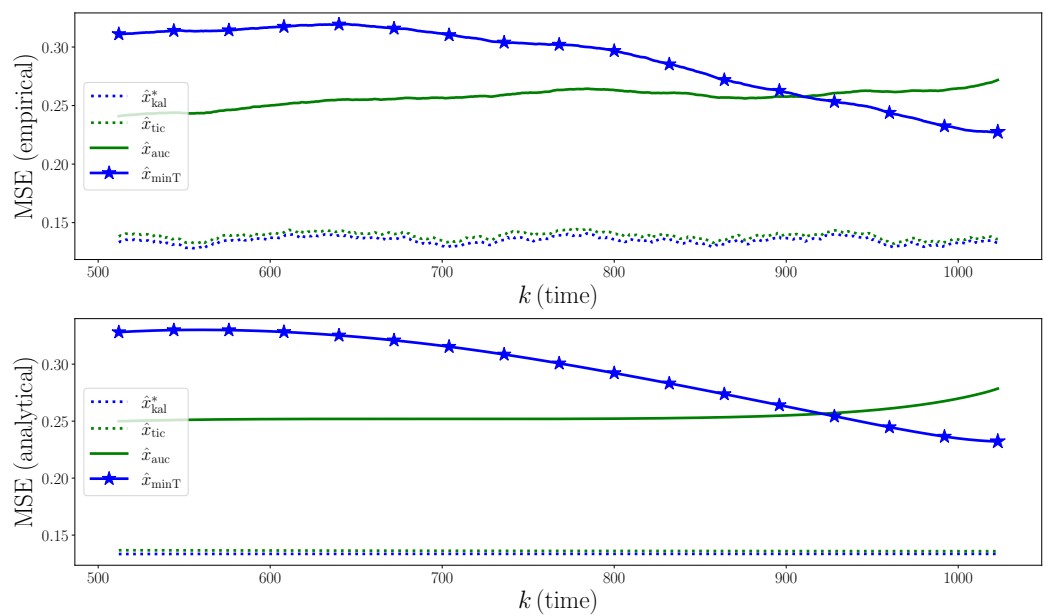

Figure 9: **MSE distortion on Coupled inverted pendulums for perceptual and non-perceptual filters (near the time $T$).** $\hat{x}_{\mathrm{auc}}, \hat{x}_{\mathrm{minT}}$ are PKF outputs minimizing different objectives. Observe that while both possess perfect-perceptual quality, they yield different estimations. Also, pay attention to the MSE gap between the MSE-optimal, but not perceptual, Kalman filter and the PKFs.

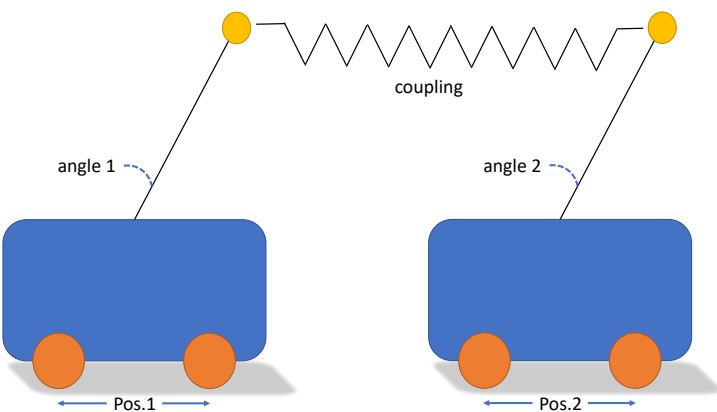

Figure 8: **Coupled inverted pendulums.**

We simulate the system for $2^{10}$ time steps ($T = 2^{10} - 1$), over $N = 2^{10}$ independent experiments. In Figures 9 and 10 we show the MSE distortion as a function of time, $\mathbb{E}\left[\|\hat{x}_k - x_k\|^2\right]$, for the different filters of Table 3; $\hat{x}^*_{\mathrm{kal}}$ is the optimal Kalman filter. $\hat{x}_{\mathrm{tic}}$ is the perceptual filter without consistency constraints, given in (10). $\hat{x}_{\mathrm{auc}}$ is the PKF output minimizing the total cost (130). $\hat{x}_{\mathrm{minT}}$ (marked by '$\star$') is the PKF output minimizing the terminal cost (129).

We observe that filters satisfying the perfect perceptual quality constraint ($\hat{x}_{\mathrm{auc}}$ and $\hat{x}_{\mathrm{minT}}$) achieve higher distortions compared to the per-sample only perceptual filter $\hat{x}_{\mathrm{tic}}$, which in turn attains MSE distortion slightly higher than that of the MSE-optimal Kalman filter. This demonstrates again the cost of temporal consistency in online estimation. Note also that PKFs minimizing different cumulative objectives, yield different estimations; while $\hat{x}_{minT}$ is optimal at termination time $T$, $\hat{x}_{\mathrm{auc}}$ achieves a lower MSE on average. As we will see next, both filters attain the same perceptual quality.

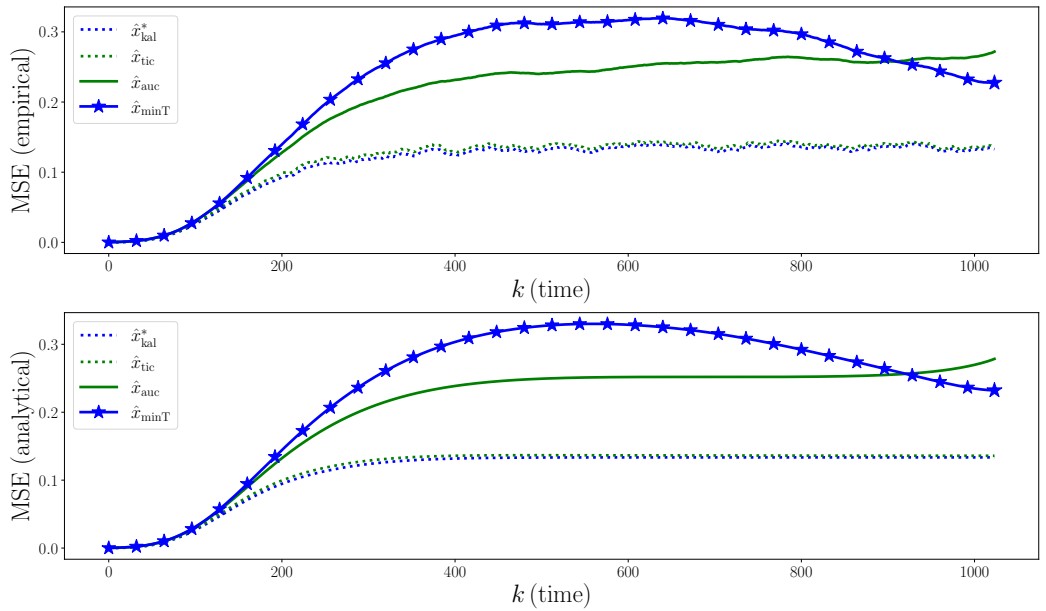

Figure 10: **MSE distortion on Coupled inverted pendulums for perceptual and non-perceptual filters (full view).**

In Fig. 11 we estimate the perceptual quality, given by the Wasserstein distance between the ground-truth distribution and the empirical Gaussian distributions of the different filters outputs. In Fig. 11(top) we estimate the distance between single-sample distributions, while in Fig. 11(bottom) we consider the joint distributions of 16 state-vectors, $x_t, t \in [k-15, k]$. Observe that while each sample of $\hat{x}_{\text{tic}}$ is distributed similarly to its reference sample, it fails to attain perfect perceptual quality where we measure the distance from the real process distribution. PKF outputs attain low perceptual index (high quality) in both scenarios. We also present the perceptual quality measured for the ground-truth signal $x_{\text{gt}}$ empirical distribution, as a reference.

Figure 12 shows the asymptotic behavior (empirical error for large horizon $T$) of $\hat{x}_{stat.}$, the stationary PKL (125). The figure also presents the empirical errors for Kalman filter and its stationary version (multiplied by a factor of 2, which is an upper bound on the MSE distortion of perceptual estimators without temporal constraints, see [3]), and the theoretical steady-state error of (125), obtained by optimizing (128) (dashed horizontal line) for comparison. The error of the non-stationary perceptual filter $\hat{x}_{\text{auc}}$ is also shown.

### I.3 Dynamic texture

Here we illustrate the qualitative effects of perceptual (temporally consistent) estimation in a simplified video restoration setting. Please see the supplementary video for the full videos. This setup visually demonstrates how:

1. Filters with no perfect perceptual quality tend to generate non-realistic images or atypical motion (random or slow movement, flickering artifacts etc.).

2. PKF outputs are natural to the domain, both spatially and temporally.

For this extent, we introduce the 'Dynamic Texture' domain. In this domain, video frames are generated from a latent state which represents their *Factor-Analysis* (FA) decomposition (see *e.g.* Bishop and Nasrabadi [2, Sec. 12.2.4] for more details). The dynamics in the FA domain are assumed to be linear, with a small Gaussian perturbation,

$$x_k^{\text{FA}} = A^{\text{FA}} x_{k-1}^{\text{FA}} + q_k, \quad x_0^{\text{FA}} \sim \mathcal{N}(0, I), \quad x_k^{\text{FA}} \in \mathbb{R}^{128}. \tag{144}$$

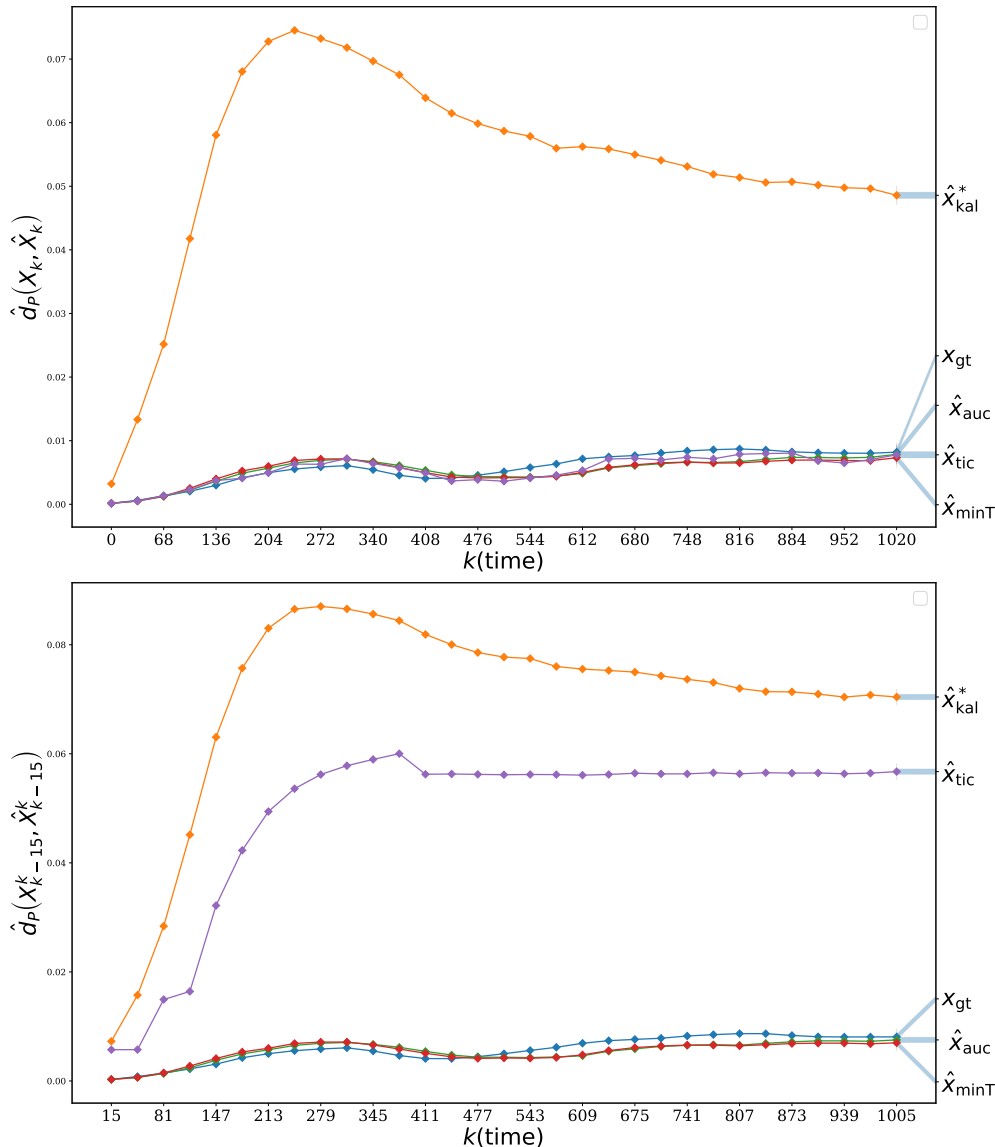

Figure 11: **Perceptual quality measured by estimated Wasserstein distance** $\hat{d}_P$ **(lower is better).** **(top)** Distance between distributions of single samples $p_{x_k}$ and $p_{\hat{x}_k}$. **(bottom)** Distance between distributions of 16-state vectors (at times $[k-15, k]$), $p_{X^k_{k-15}}$ and $p_{\hat{X}^k_{k-15}}$. Observe that $\hat{x}_{\text{tic}}$ single samples are distributed similarly to the ground-truth signal, but they fail to attain the reference joint distribution between timesteps. PKF outputs $\hat{x}_{\text{auc}}$ and $\hat{x}_{\text{minT}}$ attain high measured quality in both cases.

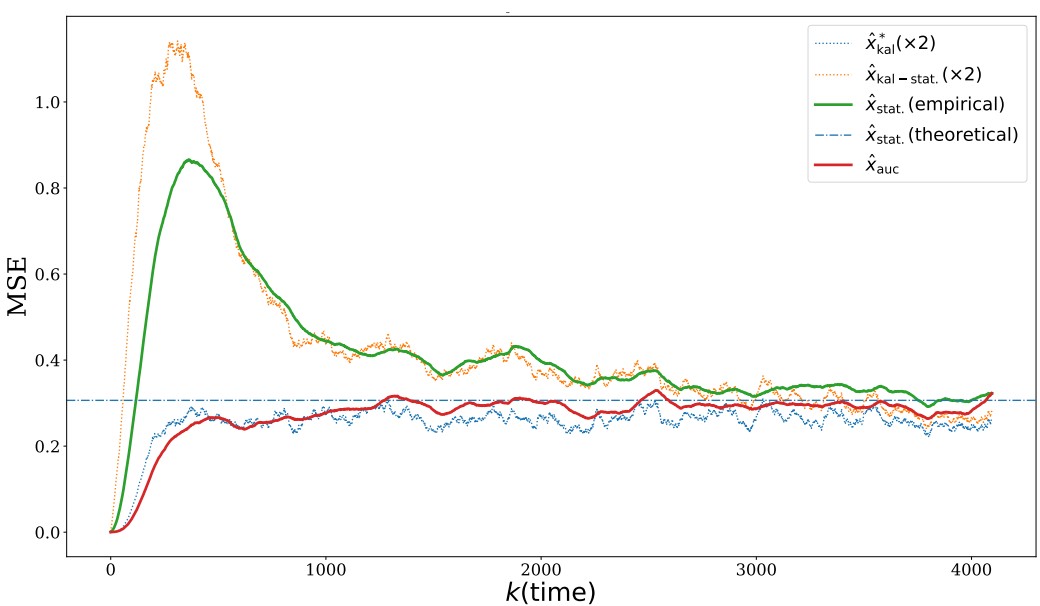

Figure 12: **MSE distortion on Coupled inverted pendulums for stationary filters.**

The state vector with the given dynamics creates frames of a wavy lake in the video domain [3], through an affine transformation,

$$x_k^{vid} = W_{\text{FA} \to vid} \left( x_k^{\text{FA}} + \varepsilon^{\text{FA}} \right). \tag{145}$$

$W_{\text{FA} \to vid}$ is a linear transformation from $\mathbb{R}^{128}$ latent states to $\mathbb{R}^{512 \times 512 \times 3}$ frames, and $\varepsilon^{\text{FA}}$ is a constant vector. $A^{FA}$ and the noise $q_k$ parameters are estimated similarly to [5]. Linear observations $y_k \in \mathbb{R}^{32 \times 32}$ are given in the frame (pixel) domain, by

$$y_k = C_k x_k^{\text{FA}} + r_k. \tag{146}$$

At times where information is being observed,

$$C_k = C_{\times 16} W_{RGB \to y} W_{\text{FA} \to vid}, \tag{147}$$

where $W_{RGB \to y}$ is a projection onto the $Y$-channel (grayscale) and $C_{\times 16}$ is a matrix that performs $16\times$ downsampling in both axes. At times where there is no observed information, $C_k = 0$. Here, $r_k$ is a Gaussian noise.

In our first experiment, measurements are supplied as in (147) up to frame $k = 127$ and then vanish ($C_k = 0, k \geq 128$), letting the different filters predict the next, unobserved, frames of the sequence. We pass $y_k$ as an input to the various filters (see Table 3); $\hat{x}_{\text{kal}}^*$ is the Kalman filter output. $\hat{x}_{\text{tic}}$ is the perceptual filter in the spatial domain, given in (10). $\hat{x}_{\text{auc}}$ is our Algorithm (PKF) output reducing the total cost in the latent space, $\mathcal{C}_{\text{auc}} = \sum_{k=0}^{T} \mathbb{E} \left[ \| x_k^{\text{FA}} - \hat{x}_k \|^2 \right]$. All filtering is done in the latent domain, and then transformed to the pixel domain. MSE is also calculated in the FA domain. In (Fig. 13) we can see that until frame $k = 127$, all filters reconstruct the reference frames well. Starting at time $k = 128$, when measurements disappear, we observe that the Kalman filter slowly fades into a static, blurry output which is the average frame value in this setting. This is definitely a non-'realistic' video; Neither the individual frames nor the static behavior are natural to the domain. Our perfect-perceptual filter, $\hat{x}_{\text{auc}}$, keeps generating a 'natural' video, both spatially and temporally. This makes its MSE grow faster.

We now perform a second experiment, where $C_k$ is set to zero until frame $k = 512$. At times $k \geq 513$ measurements are given again by the noisy, downsampled frames as described in (146)-(147). In Fig. 14 we present the outcomes of the different filters. We first observe that up to frame $k = 512$, there is no observed information, hence outputs are actually being generated according to priors.

---

[3]Original frames are taken from 'river-14205' by OjasweinGuptaOJG via pixabay.com, and are free to use under the content licence.

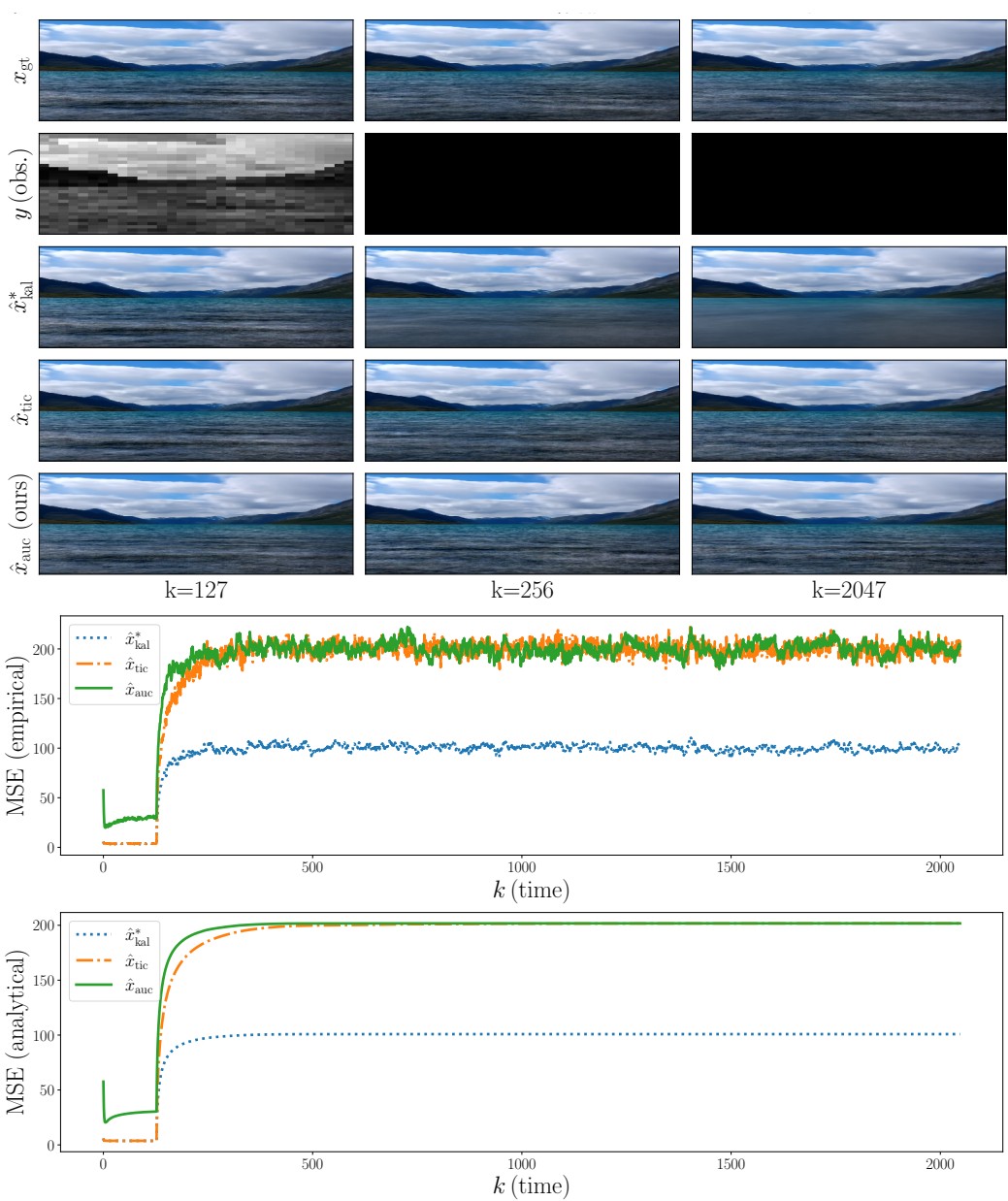

Figure 13: **Frame prediction on a dynamic texture domain.** In this experiment, measurements are supplied only up to frame $k = 127$. The filter's task here is to predict the unobserved future frames of the sequence. Observe that the $\hat{x}^*_{\text{kal}}$ fades into a blurred average frame, while the perceptual filter $\hat{x}_{\text{auc}}$ generates a natural video, both spatially and temporally. This makes its MSE grow faster,

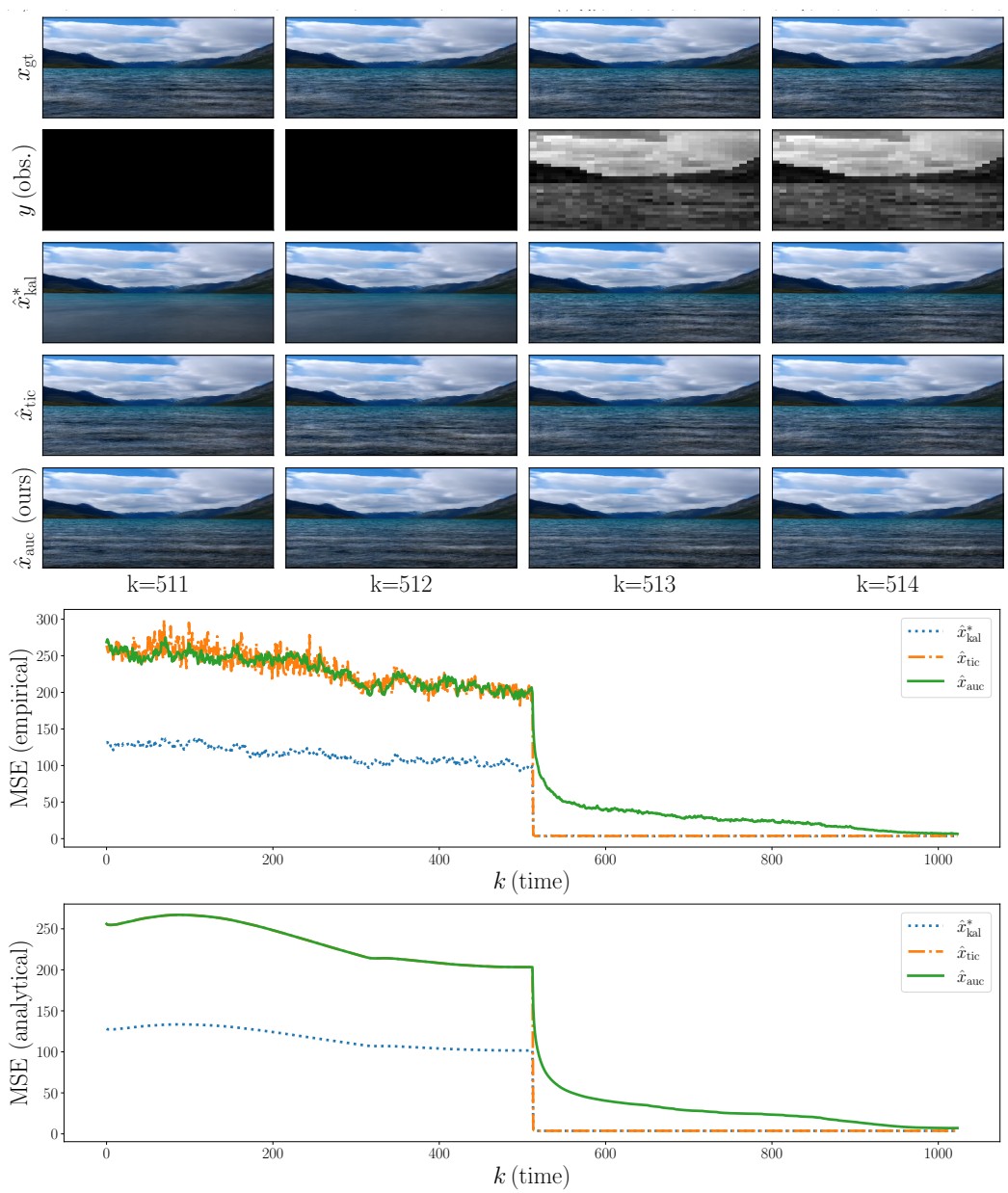

Figure 14: **Frame generation on Dynamic texture domain.** In the first half of the demo ($k \leq 512$), there are no observations, hence the reference signal is restored according to prior distribution. We observe that filters with no perfect-perceptual quality constraint in the temporal domain generate non-realistic frames (Kalman filter output $\hat{x}^*_{\text{kal}}$) or unnatural motion ($\hat{x}_{\text{tic}}$). Perceptual filter $\hat{x}_{\text{auc}}$ is constrained by previously generated frames and the natural dynamics of the domain, hence its MSE decays slower.

The Kalman filter outputs a static, average frame. $\hat{x}_{\text{tic}}$ randomizes each frame independently, which creates the impression of rapid, random movement with flickering features, which is unnatural to the reference domain. At frame $k = 513$, when observations become available, we can see that $\hat{x}^*_{\text{kal}}$ and $\hat{x}_{\text{tic}}$ are being updated immediately, creating an inconsistent, non-smooth motion between frames 512 and 513. PKF output $\hat{x}_{\text{auc}}$, on the other hand, keeps maintaining a smooth motion. Since non-consistent filters outputs rapidly becomes similar to the ground-truth, their errors drop. The perfect-perceptual filter, $\hat{x}_{\text{auc}}$, remains consistent with its previously generated frames and the natural dynamics of the model, hence its error decays more slowly.

