# OpenReview forum: "Perceptual Kalman Filters: Online State Estimation under a Perfect Perceptual-Quality Constraint"
_NeurIPS.cc/2023/Conference — NeurIPS 2023 poster_

### Official Review · Reviewer_c5sG · 2023-06-20

**Soundness:** 4 excellent
**Presentation:** 3 good
**Contribution:** 3 good
**Rating:** 6
**Confidence:** 3

**Summary:**

This work studies the problem of temporal singal reconstruction for corrupted data. Under a perfect perceptual-quality constraint, the previously regarded optimal model, i.e, the Kalman filter is shown to confronted with a fundamental dilemma. A recursive formula for perceptual filters is proposed and be empirically validated on a video reconstruction problem.


**Strengths:**

+ This work is put in a proper context in the liteature, where the perception-distortion trade-off is an important theory that only attract few research interests.
+ Reasonable problem formulation, and detailed theoretical drivations.


**Weaknesses:**

- Is the proposed tool generalizable to other fidelity measures in addition to MSE?
- The effecrtiveness of the proposed tool is only validated on a single sample, more are expected.



**Questions:**

It's better to split Eq. 17 into multiple lines for improved formatting.

---

> ### Author Rebuttal · Authors · 2023-08-08
>
>
> *Is the method can be generalized beyond MSE?*
>
> Please note that in Thm. 4.1 we have a general form for linear perfect-perceptual quality filters (Eq.14). While the form (14) is optimal for MSE, it can be used as a representation for linear filters in general (but see restriction below).
> The constraints on the coefficients ($Q_k-\pi_k S_k \pi_k^\top - \phi_k \Sigma \phi_k^\top \succeq 0$) are necessary and sufficient for perfect perception, again, regardless of the cost objective. It indeed might be possible to optimize coefficients for objectives other than MSE under these constraints to gain optimal perceptual *linear* filters. Note, however, that under general distortion measures, optimal filters might be non-linear. We will add this discussion to the paper. Thank you for the interesting remark.
>
> *More validation is expected:*
>
> We validate our method on a 2d oscillator example, and two video reconstruction scenarios, which were included in the main text within the space limitations. Additional experimental results can be found in  Appendix H, including inverted pendulums and further results for the video reconstruction.
>
> Please refer to our "global" response, where we add an experiment for demonstrating many additional different dynamics. We will add this demonstration to the final version.
>
> *Formatting Eq. 17:*
>
> Thank you for your suggestions, we will format eq. 17 as you suggested,

---

> > ### Comment · Reviewer_c5sG · 2023-08-12
> > **Post-rebuttal**
> >
> > The reviewer appreciates the authors' responses to address the raised concerns.

---

### Official Review · Reviewer_ZP1L · 2023-07-05

**Soundness:** 3 good
**Presentation:** 4 excellent
**Contribution:** 3 good
**Rating:** 7
**Confidence:** 2

**Summary:**

This article addresses the problem of optimal causal filtering under a perfect perceptual-quality constraint. The authors introduce the concept of an unutilized information process and present a recursive formula for perceptual filters. The study demonstrates the effects of perfect perceptual-quality estimation on video reconstruction. Overall, this article contributes to understanding optimal causal filtering under a perceptual-quality constraint and offers new insights for addressing this issue.

**Strengths:**

Unfortunately, for me the paper was not easily accessible as I am not familiar with studies of the signal reconstruction.
·The overall problem is interesting and has practical significance.
·I find this work quite original to the best of my limited knowledge.
·The paper to me seems technically sound and theoretical assumptions are formally proved.
·The appendix materials have been carefully prepared, which can be of great help in understanding the paper. Meanwhile, the open-source code provides convenience for community reference.

**Weaknesses:**

·Experimental analysis is limited, is it necessary to showcase more diverse video scenarios?
·Is it necessary to add a conclusion section to summarize the conclusion of this paper and clarify its possible application scenarios, as well as the interests can be followed in the future research.

**Questions:**

As mentioned above.

**Limitations:**

The paper clearly describes societal impact and potential limitations of their work.

---

> ### Author Rebuttal · Authors · 2023-08-08
>
> Thank you for your positive response.
>
> Our focus in this work was on analytic closed form results. To achieve this we applied our algorithms and analysis to the Gauss-Markov setting, where we also focus our empirical efforts. Future work, extending our results to more general domains, will be able to compare empirical results in more diverse settings. In this respect, please also see our response to reviewer 3Cww.
>
> We did not include a conclusion section due to the limited space. We will add it to the final version.
>
> Pay attention that additional experimental results can be found in the Appendix, as well as in our "global" response..

---

### Official Review · Reviewer_3Cww · 2023-07-06

**Soundness:** 3 good
**Presentation:** 2 fair
**Contribution:** 2 fair
**Rating:** 4
**Confidence:** 4

**Summary:**

This paper aims to study the problem of optimal causal filtering under a perceptual-quality constraint. The main contribution of this paper is to provide a mathematical framework and a closed-form solution to the aforementioned problem under some mild conditions. The experimental results on a video reconstruction problem demonstrate the effectiveness of perceptual-quality estimation.

**Strengths:**

1. The cost of temporal consistency investigated in this study holds significant importance in the field of online restoration.
2. The experimental results on a video reconstruction problem provide evidence for the qualitative effects of the proposed perceptual filtering.

**Weaknesses:**

1.	The readability of the paper is restricted due to the oversimplification of the formula derivation, such as in the case of Formula 16.

**Questions:**

1. The authors should concisely explain why they did not compare their method with other perceptual-quality constrained SOTA methods in online restoration. Adding a comparison with SOTA methods, rather than solely comparing to the Kalman filter, would enhance the overall persuasiveness.

2. When both constraints 6 and 7 are applied simultaneously, it may result in an additional MSE loss. Under what conditions can this loss be alleviated while still satisfying both constraints?

**Limitations:**

No limitations are addressed in this paper. It is recommended that the authors make an effort to improve the readability of the paper.

---

> ### Author Rebuttal · Authors · 2023-08-08
>
> Thank you for your comments.
>
> We did our best to make our work readable. We will make an effort to clarify equation derivations and improve readability in the final version.
> As for Eq.16, it is a direct consequence of the MMSE orthogonality property, see e.g. [8]. We will clarify this in the text.
>
> To the best of our knowledge, there is no other work that addresses perfect-perception reconstruction (in the mathematical sense) in causal filtering problems. While temporal consistency is an emerging topic in the design of deep models and architectures, none of the existing methods are designed to achieve perfect perception as in our work. We will add a discussion on this issue to the paper.
>
> Under the causality constraint (6), the deterministic Kalman filter is known to be MSE optimal. Now considering (7), the MSE is expected to grow, as shown analytically in the static case in [8]. This cost is alleviated in the extremely degenerate cases, where KF perfectly estimates the trajectory (e.g. observations are full or process is deterministic).

---

> > ### Comment · Reviewer_3Cww · 2023-08-22
> >
> > Thanks for the response. After reading the response as well as the other reviewers' comments, I would like to increase my rating.

---

### Official Review · Reviewer_j8n4 · 2023-07-07

**Soundness:** 4 excellent
**Presentation:** 4 excellent
**Contribution:** 3 good
**Rating:** 6
**Confidence:** 4

**Summary:**

This submission introduces the Perceptual Kalman filter as a tractable solution to the online state estimation problem under perfect perpetual-quality constraints. This means the authors review the problem of state estimation under perceptual constrains (i.e. the joint distribution of any sub-sequence of states should match the original data). Then they derive a tractable approximation of the perceptual filters which have an analytic solution. Experiments were conducted on a harmonic oscillator and a dynamic texture domain. The harmonic oscillator experiments capture the MSE due to perceptual consistency and also due to the approximation used for tractability. The dynamic prediction domain demonstrates the approach scales to high-dimensions.

**Strengths:**

1. Clear and thorough presentation of perceptual estimation and its relationship to prior work (in particular the Kalman Filter).
2. Easy-to-understand derivation of the Perceptual Kalman filter and in particular the trade-offs made in order to obtain tractability.
3. Experiments that clearly compare how the technique compares to prior work, the theoretical maximum performance and the cost of tractability. Scalability experiments also demonstrate that this works for large linear systems.

**Weaknesses:**

1. Although the linear systems and observations have wide applicability, the paper would be improved by mentioning this limitation and why it has or doesn't have a significant effect in the applications where perceptual quality of state-estimates is desired. This is mainly to explain how widely the linear results presented by the paper apply to the problem landscape of the general case.
2. The experiments only show results for a harmonic oscillator. Although this is extremely useful as away to visualize the consistency gap and the gap due to $\Phi_k = 0$, experiments showing how different dynamics can change these gaps would be very useful.

**Questions:**

How bad can the $\Phi_k = 0$ approximation hurt your performance? Is there some provable bound for the worst-case?

**Limitations:**

I think the authors present the limitations of the method clearly.

---

> ### Author Rebuttal · Authors · 2023-08-08
>
> Thank you for your comments.
>
> Nonlinear filtering is in general analytically intractable even in the standard setting of classic state estimation without constraints. Given the absence of current theoretical analysis of perceptual constraints in filtering, it is natural to focus on the linear setting at the current state of knowledge and gain analytical insight. Even this has proved to be challenging, and a clear goal is to extend this work to more general settings. While temporal consistency is an emerging topic in the pracical design of deep models and architectures, none of the existing methods are designed to achieve perfect perception as in our work, hence empirical comparisons are hard. We will add a discussion on this issue to the paper.
>
> An interesting direction for future work can be a study of domains where there are hidden linear-Gauss dynamics in some latent space (e.g. using GAN, VAE etc.) while transformation back to signal space is non linear. We will mention this as a future work
>
> Regarding a bound on the degradation due to $\varPhi_k=0$ reduction: In fact, there is no such general bound. There are possible cases where this reduction may lead to worse error, while in other scenarios it might be optimal. Let us examine two extreme cases. In one extreme, when an observation is missing, the corresponding innovation is uncorrelated with the estimand, hence discarding past unutilized information means that the state update step in Algorithm 1 is completely random, which results in high MSE. In the other extreme, for processes where states at different timesteps are weakly correlated ($\rho(A)\ll 1$ or $\rho(Q) \gg \rho(P_0))$, unutilized information rapidly becomes irrelevant and can be safely discarded.
>
> We empirically demonstrate the efficiency of the reduced filters in our numerical Section (Fig. 3) and in Appendix H.2. We’ll add a discussion about the intuition behind the influence of  $\varPhi_k$. Thank you for this interesting remark.
>
> Regarding your request to show gaps for different dynamics. We conducted an experiment that demonstrates the MSE gaps for different settings. The experimental details and numerical results appear in our "global" response and related PDF. We will add this demonstration to the final version.

---

### Author Rebuttal · Authors · 2023-08-08

We thank all reviewers for their comments.

Following some of your remarks, we conducted an additional experiment to demonstrate different MSE gaps between temporally consistent and inconsistent filters (based on harmonic oscillators with different dynamics). Detailed description and results appear in the attached PDF.

Specifically, this experiment demonstrates how different dynamics yield different gaps (due to temporal consistency, and due to $\varPhi_k = 0$). We used different harmonic oscillator dynamics (Section 5.1), where the marginally stable matrix $A$ is multiplied by a varying factor of $\rho$, and observations are given only up to a certain time.
We plot the terminal mse, at time $T$ for each filter, normalized by the variance of $x_T$. The estimator $\hat X_{direct} $ is established using the direct optimization approach (Appendix C).

We observe that when timesteps are more correlated, both consistency and unutilized info play a major role, hence MSE gaps between filters get bigger. This experiment also demonstrates that past unutilized information is helpful in the absence of current obsetvations.

We will add this experiment to the paper.

---

### Decision · Program_Chairs · 2023-09-21

**Decision:**

Accept (poster)

**Comment:**

Reviewers and the AC read the rebuttal and took that into consideration for their final recommendation. All reviewers participated in the discussion and their initial concerns were adequately addressed. Several suggestions for improving the exposition were made, which should be incorporated into the camera-ready paper.